# TRIM28 modulates nuclear receptor signaling to regulate uterine function

Rong Li[1], Tianyuan Wang[2], Ryan M. Marquardt ®[1], John P. Lydon[3], San-Pin Wu ®[1] & Francesco J. DeMayo ®[1] ✉

Estrogen and progesterone, acting through their cognate receptors the estrogen receptor α (ERα) and the progesterone receptor (PR) respectively, regulate uterine biology. Using rapid immunoprecipitation and mass spectrometry (RIME) and co-immunoprecipitation, we identified TRIM28 (Tripartite motif containing 28) as a protein which complexes with ERα and PR in the regulation of uterine function. Impairment of TRIM28 expression results in the inability of the uterus to support early pregnancy through altered PR and ERα action in the uterine epithelium and stroma by suppressing PR and ERα chromatin binding. Furthermore, TRIM28 ablation in PR-expressing uterine cells results in the enrichment of a subset of TRIM28 positive and PR negative pericytes and epithelial cells with progenitor potential. In summary, our study reveals the important roles of TRIM28 in regulating endometrial cell composition and function in women, and also implies its critical functions in other hormone regulated systems.

Both estrogen and progesterone exert pleiotropic roles not only in reproductive biology but also in cardiovascular, metabolic and immune physiological processes[1,2]. In addition to their important physiological functions, estrogen and progesterone have also been associated with a multitude of pathological conditions, including gynecological, metabolic and cardiovascular disorders, and cancers[1–3].

Uterine function is tightly regulated by estrogen and progesterone signaling through their nuclear receptors, the estrogen receptor α (ERα) and progesterone receptor (PR) respectively. These receptors regulate endometrial cell proliferation, differentiation and the ability of the endometrium to support embryo implantation and fetal development[4,5]. Perturbation in the homeostatic levels of either nuclear receptor causes uterine dysfunction that leads to fertility disorders[2,6]. While co-regulators such as members of the steroid receptor coactivator family have been shown to act with ERα and PR[7,8], the modulators of PR and ERα mediated transcription during decidualization remain largely unknown.

TRIM28 is a ubiquitously expressed multi-functional protein[9]. It plays a crucial role in the maintenance of pluripotency and epigenetic stability in human stem cells and mouse germ cells acted as the transcriptional repressor[10]. In the adult, TRIM28 has been shown to regulate T cell activation[11], muscular growth[12], and adipocyte adipogenesis[13]. Although the molecular mechanism of TRIM28 in the differentiated tissues is not fully understood. The present study links TRIM28 with the steroid hormone receptors ERα and PR in regulating uterine biology which also provides the insight of TRIM28 functions in other hormone regulated systems, including the adipocyte, muscle and immune systems, etc.

## Results

### TRIM28 interacts with PR in decidual human endometrial stromal cells (HESCs)

PR is the central regulator of decidualization[6]. Rapid immunoprecipitation mass spectrometry of endogenous proteins (RIME)[14] was performed to identify the PR containing chromatin complex of decidualized primary human endometrial stromal cells (HESCs). Top candidates for PR partner proteins were determined based on interaction probability (Fig. 1a). Among them, we focused on TRIM28, a multi-functional transcription factor[9], which is ubiquitously expressed in the nuclei of pre-decidual and decidual HESCs (Fig. 1b). In a subset of decidualized HESCs,

[1]Reproductive and Developmental Biology Laboratory, National Institute of Environmental Health Sciences, Research Triangle Park, NC, USA. [2]Integrative Bioinformatics, National Institute of Environmental Health Sciences, Research Triangle Park, NC, USA. [3]Department of Molecular and Cellular Biology, Baylor College of Medicine, Houston, TX, USA. ✉e-mail: demayofj@niehs.nih.gov

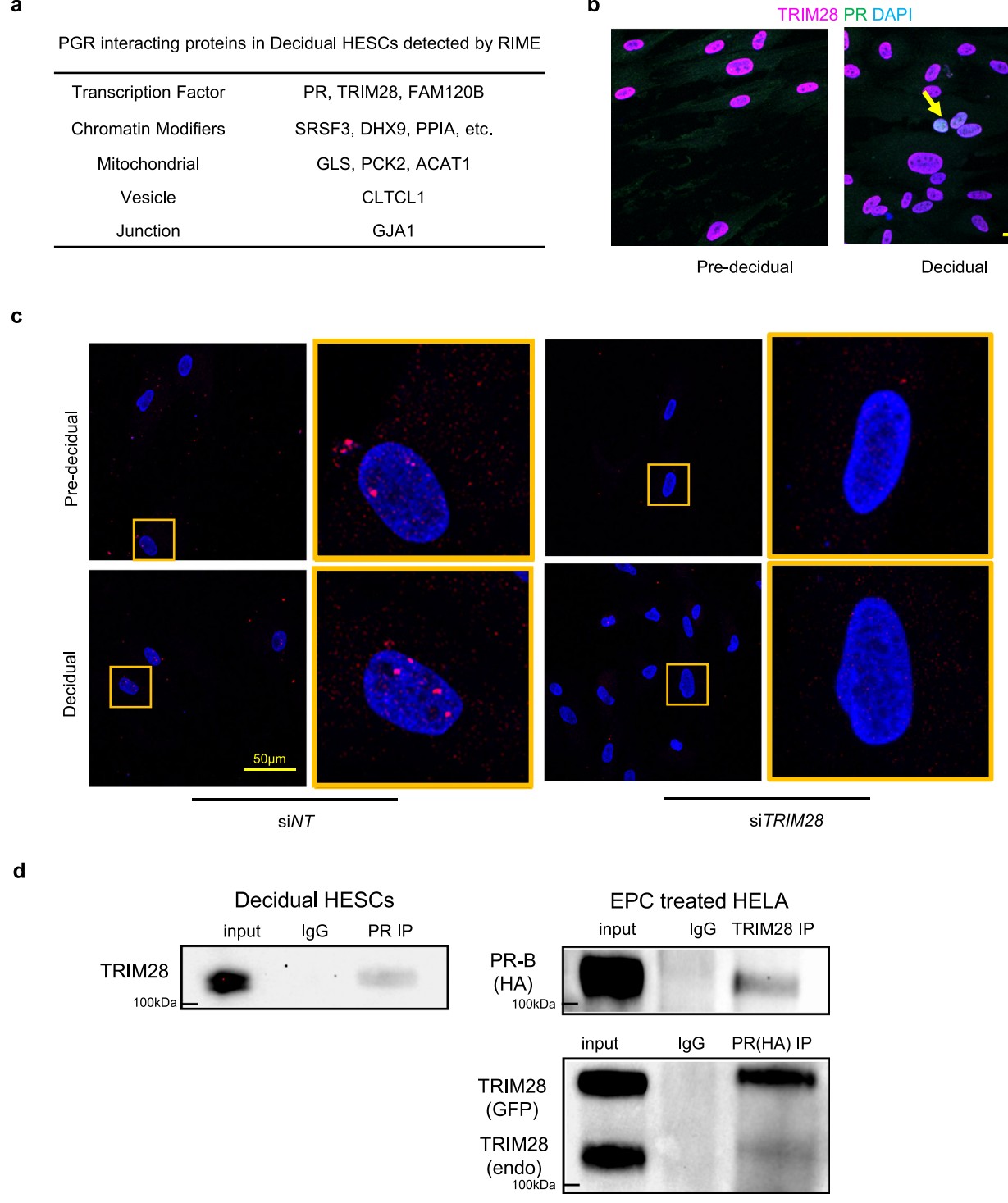

**Fig. 1 | PGR interacts with TRIM28 to regulate decidualization in HESCs. a** PR interacting proteins in decidual HESCs by RIME. **b** Immunofluorescence (IF) of TRIM28 (magenta) and PR (green) in pre-decidual and decidual HESCs. **c** TRIM28 and PR proximity (red dots, arrow pointed) in si*NT* and si*TRIM28* treated pre-decidual and decidual HESCs by Proximity Ligation Assay.

**d** Co-immunoprecipitation of TRIM28 and PR in the decidual HESCs and PR/TRIM28 overexpressed HELA cells. The IF was repeated in three primary cell line. The proximity and co-IP in decidual HESCs were repeated in two primary cell lines. The co-IP in the overexpressed HELA cells was repeated for three independent experiments. All repeats show similar results.

TRIM28 and PR were co-expressed in the nucleus in close proximity (Fig. 1b, c). Moreover, the immunoprecipitation (IP) assay using the PR antibody successfully pulled down TRIM28 in decidual HESCs further validated the RIME results. Co-IP of PR and TRIM28 were also accomplished in the HA tagged PR and GFP tagged TRIM28 overexpressed HELA cells (Fig. 1d). These results demonstrated the close proximity of TRIM28 within the PR chromatin complex.

## TRIM28 regulates decidualization of HESCs
The function of TRIM28 in HESCs was assessed by siRNA-mediated gene silencing. Knocking down of TRIM28 decreased cell proliferation and migration but did not affect HESC apoptosis (Fig. 2a, Supplementary Fig. 1a–c). Additionally, si*TRIM28* treated HESCs failed to decidualize in which the spindle shaped fibroblast did not transform into the rounded epithelioid cells following hormone stimulation

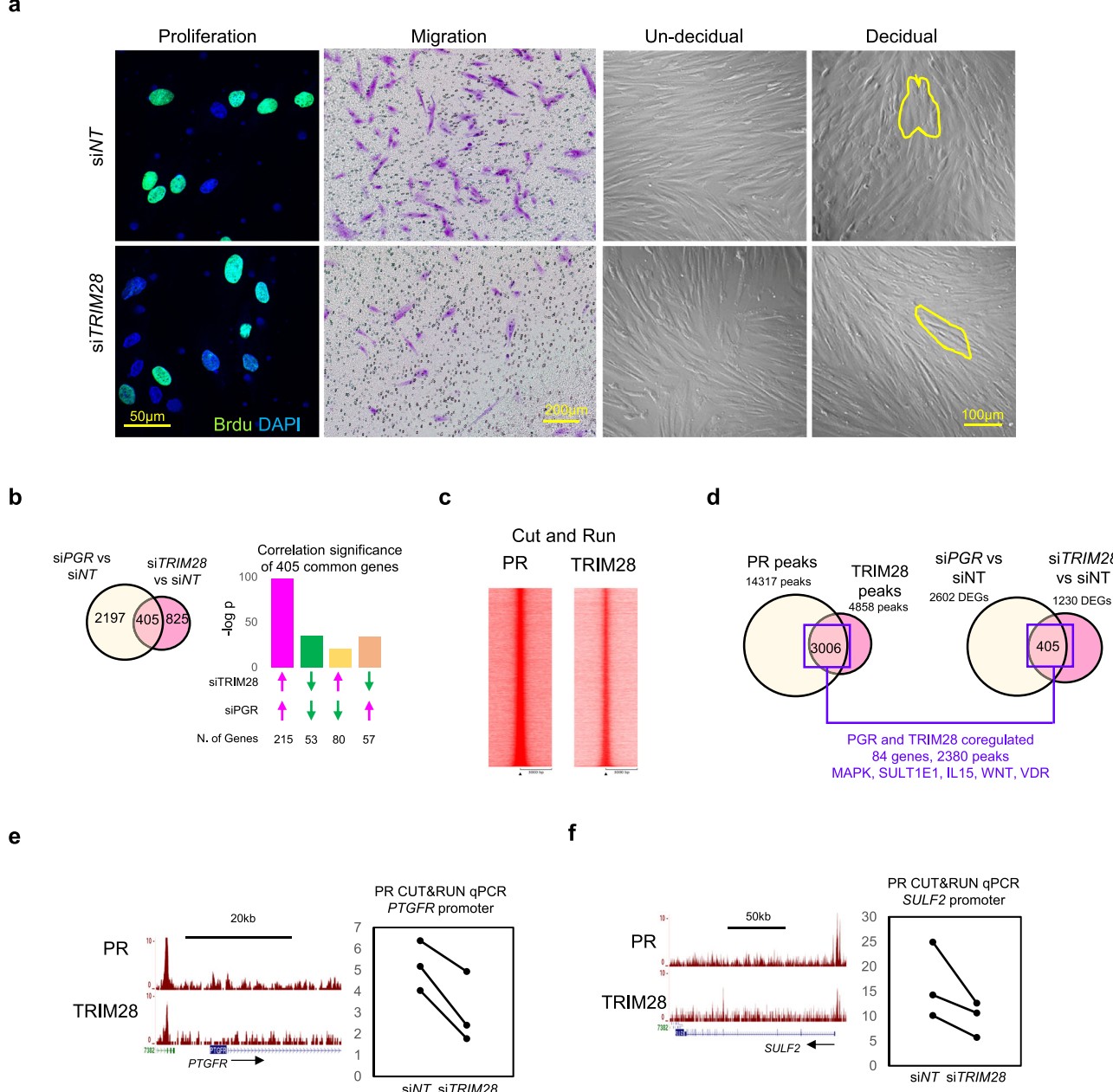

**Fig. 2 | TRIM28 knockdown disrupt PR signaling in HESCs. a** Edu labeled cell proliferation, migrated cells in the transwell assay, and brightfield images of pre-decidual and decidual HESCs. Yellow line circles the cells. **b** Comparison of si*PGR* and si*TRIM28* mediated transcriptomic changes in the hormone stimulated decidual HESCs. **c** The heatmap of PR and TRIM28 binding peaks in decidual HESCs by CUT&RUN. **d** The overlapped peaks of TRIM28 and PR, the associated common DEGs altered by *TRIM28* and *PGR* knockdown in the decidual HESCs. Genome browser display of PR and TRIM28 binding peaks, and the CUT&RUN PCR of PR binding activity at *PTGFR* **e** and *SULF2* **f** promoter in the si*NT* or si*TRIM28* treated decidual HESCs. *N* = 500,000 cells examined over three independent experiments and repeated in two primary cell lines with similar results. Two-sided student's t test. The bight image, cell proliferation, and proliferation were repeated in three primary cell lines with similar results.

(Fig. 2a). Expression of decidual markers *PRL* and *IGFBP1* was also lower in TRIM28-silenced HESCs compared to the control (Supplementary Fig. 1d). Both findings indicated an impairment of the decidualization process as a result of TRIM28 deficiency. The transcriptomic analysis of si*TRIM28* and si*NT* HESCs under hormone stimulation identified multiple decidual related pathways[15] were altered by *TRIM28* deficiency (Supplementary Fig. 1e, Supplementary Data 1). Among them, the predicted activity of medroxyprogesterone acetate (MPA, synthetic progesterone) was lower in the si*TRIM28* group, suggesting suppressed progesterone signaling. By comparing the gene signatures of *PGR*[16] and *TRIM28* in hormone stimulated HESCs further demonstrated a positive correlation between TRIM28 and PR in the regulation of gene

expression (Fig. 2B). Collectively, our data shows a pivotal role of TRIM28 in progesterone signaling mediated decidualization.

Given that TRIM28 and PR were closely associated protein complexes, genome occupancy patterns between TRIM28 and PR were compared in hormone stimulated HESCs using CUT&RUN. PR binding was observed in 61.9% (3006/4858) of TRIM28 occupied genomic regions (Fig. 2c, Supplementary Fig. 1f). Among the TRIM28 and PR co-occupied sites, nearly 80% (2380/3006) were predicted as cis-regulatory regions in 84 genes that are regulated by both si*TRIM28* and si*PGR* (Fig. 2d). Gene ontology analysis on these 84 genes revealed significant enrichment of functional terms for kinases (i.e. MAPK), sumoylation (i.e. SULT1E1), inflammation (i.e. IL15), and WNTs (Fig. 2d),

which are known as important biological processes for decidualization[17–19]. Among the PR-TRIM28 co-regulated genes, Prostaglandin F Receptor (*PTGFR*), sulfase 2 (*SULF2*) was decreased by si*TRIM28* treatment, and the binding activity of PR was moderately decreased at the *SULF2* promoter in the si*TRIM28* compared to the si*NT* treated groups (Fig. 2e, f). These findings indicate that TRIM28 and PR directly co-regulate a core set of genes crucial for the hormone-dependent decidualization process which were diminished by TRIM28 deficiency.

## TRIM28 regulates HESCs function prior to decidualization
Transcriptomic profiling revealed that TRIM28 pre-determines the expression of genes for HESC's decidualization capacity at the fibroblast stage, as evidenced by the high similarity on TRIM28 gene signatures between hormone-treated and untreated cells (Supplementary Fig. 1g, Supplementary data 1). This finding is further supported by the conserved pathways in both gene signatures including TGFβ, EGF and miRNA, etc (Supplementary Fig. 1h).

Notably, an increase of histone modifier EZH2 and a reduction of HDAC inhibitors predicted molecular activities were also seen (Supplementary Fig. 1h), linking TRIM28 to the regulation of the epigenomic landscape. Such an association was further supported by the increased chromatin accessibility both at the TRIM28 binding and non-binding chromatin loci detected by ATAC-Seq (Supplementary Fig. 1i, j, Supplementary data 2). These more accessible chromatin loci were predicted to be the cis-regulatory regions of 382/895 DEGs in which two thirds of the DEG showed increased expression, which are involved in the induction of TGF, EGF and the suppression of HDAC inhibitors and miRNA signaling.

Since TRIM28 does not have a DNA binding domain, TRIM28 likely interacts with DNA-binding proteins in the chromatin[9]. Motif enrichment analysis of the TRIM28 ChIP-Seq dataset in pre-decidual HESCs identified overrepresented DNA motifs for the TGFβ family (i.e. SMADs), the Notch family (i.e. RBPJ), and the STAT family that also manifested altered activities in response to *TRIM28* silencing (S1K). IP mass spectrometry of TRIM28 in the pre-decidual HESCs also found interactions of TRIM28 with regulators of transcription and chromatin remodeling, such as STAT and RhoA family members (Supplementary Fig. 1l). These results indicate TRIM28 can regulate HESC gene transcription through altered chromatin accessibility, interactions with STAT, NOTCH, and TGFβ proteins to modulate the biology of pre-decidual HESCs.

## Murine TRIM28 is required for endometrial decidualization
TRIM28 is expressed in all the uterine compartments in the pregnant mouse (Supplementary Fig. 2a). To study TRIM28's functions in vivo, we employed the *Pgr*[Cre] mouse[20] to ablate *Trim28* in PGR expressing cells of the mouse uterus (Supplementary Fig. 2b). *Pgr*[Cre/+]*Trim28*[f/f] (TRIM28[d/d]) females were infertile (Supplementary Fig. 3a). Heterozygous *Pgr*[Cre/+]*Trim28*[f/+] (TRIM28[+/d]) females were sub-fertile with a reduced litter size (Supplementary Fig. 3a), suggesting a haploinsufficient phenotype.

Based on our HESC studies, we hypothesized that TRIM28-deficiency would incur decidualization defects at pregnancy. We chose to examine the uterus at pregnancy day 4.5 (D4.5) and D7.5[21]. At D4.5, embryo implantation sites, evidenced by blue bands following an Evan's blue dye injection[22], were absent in TRIM28[d/d] uterus while hatched blastocysts were found in the uterus (Fig. 3a, Supplementary Fig. 3b). Moreover, embryo implantation and decidualization markers, including the nuclear expression of FOXO1, intercellular punctuate GJB2 and peri-embryonic COX2 expression[23–25] were not detected in the TRIM28[d/d] uterus (Fig. 3a). Due to embryo implantation failure at D4.5, TRIM28[d/d] mice did not contain a decidua ball at D7.5 (Fig. 3a, Supplementary Fig. 3c). In summary, these findings demonstrate a pivotal role for TRIM28 in embryo implantation and decidualization.

The artificial decidualization was used to confirm the intrinsic role of endometrial TRIM28 in stromal cell decidualization. In this model, decidualization is induced by hormone priming with oil injection at one horn[26]. The stimulated uterine horn in the TRIM28[d/d] uterus failed to increase in the size and weight and exhibited limited stromal cell proliferation and fewer decidual marker HAND2[27] positive cells (Fig. 3b, Supplementary Fig. 3d). These results demonstrated that endometrial TRIM28 loss caused TRIM28[d/d] mouse's inability to undergo decidualization. Meanwhile, the un-stimulated TRIM28[d/d] uterine horn exhibited reduced stromal but increased epithelial PR expression (Supplementary Fig. 3e), suggested a disruption of hormone responses of TRIM28[d/d] uterus that is pre-requisite for decidualization.

## TRIM28 is essential for endometrial progesterone signaling
Since current evidence points to an intrinsic dysfunction of uterine responses to hormones in TRIM28[d/d] mice, we next examined the uterine phenotypes at natural pregnancy D3.5 when hormone priming prepares the uterus for embryo implantation. No ovarian phenotypes were observed in the TRIM28[d/d] mice, despite of a higher incidence of embryos with abnormal morphology and oviductal embryo retention at D3.5 (Supplementary Fig. 3F–H). TRIM28[d/d] uteri were thinner with decreased stromal and increased epithelial proliferation and an altered vasculature structure (Fig. 3c, Supplementary Fig. 3I–K). These data again support an intrinsic role for uterine TRIM28 in early pregnancy establishment.

RNA-Seq was performed on D3.5 uteri to examine the altered uterine transcriptome in the TRIM28[d/d] mice (Fig. 3d, Supplementary data 3). Comparing the TRIM28 gene signature with those of published genetic modified mouse models that exhibited decidualization defects[28–35], TRIM28 deficiency is correlated with suppressed PR and FOXA2 signaling while enhanced ERα signaling (Fig. 3d). We further validated the decreased expression of PR target genes such as *Ihh*, *Nr2f2*, *Hand2*, *Il13ra*, *Areg*, and *Npl*, reduced gland related genes such as *Foxa2*, *Lif*, *Lifr*, *Il6st*, *Spink3*, as well as increased estrogen target genes, such as *Muc1*, *Ramp3*, *Ltf1*, *Inhbb*, *Greb1*, and *Cftr* in TRIM28[d/d] uteri (Supplementary Fig. 4A, B). Furthermore, activated inflammatory responses IFNG, Wnt-catenin and inhibited angiogenesis VEGF, hedgehog SHH, etc. pathways were also predicted to be altered in TRIM28[d/d] uteri (Supplementary Fig. 4C). These observations indicate an aberrant uterine hormone signaling milieu as a result of TRIM28 deficiency.

Given that progesterone signaling is suppressed in the TRIM28[d/d] uteri, as inferred by the mifepristone (a PR antagonist) being one of the top activated molecular activities (Supplementary Fig. 4c), we next examined the impact of TRIM28 loss on progesterone signaling. Ovariectomized *Pgr*[cre/+] mice were treated with progesterone or oil for 6 h to identify the progesterone gene signature (Fig. 3e). TRIM28 and progesterone gene signatures shared 383 genes with 78% of them changed in the opposite direction (Fig. 3e), indicating suppressed PR signaling in the TRIM28[d/d] uteri. Notably, PR and ERα protein levels were not altered in whole uterine extracts from the TRIM28[d/d] mice (Supplementary Fig. 4d) and most TRIM28[d/d] epithelial cells maintained comparable PR signal intensity as that of control animals (Fig. 3f). The unaltered epithelial PR expression in conjunction with the inhibited epithelial progesterone responsive genes demonstrate a defective transduction of progesterone signaling in the TRIM28[d/d] uterine epithelia. In contrast, an obvious reduction of the PR protein level was found in TRIM28[d/d] stromal cells (Fig. 3f) which is the major cause of suppressed stromal progesterone signaling. A similar mRNA expression profile was also found for *Pgr* (Fig. 3f).

## TRIM28 deletion decreased PR chromatin binding activities in the epithelium
Defective transduction of progesterone signaling in the epithelial compartment was further supported from observations that the

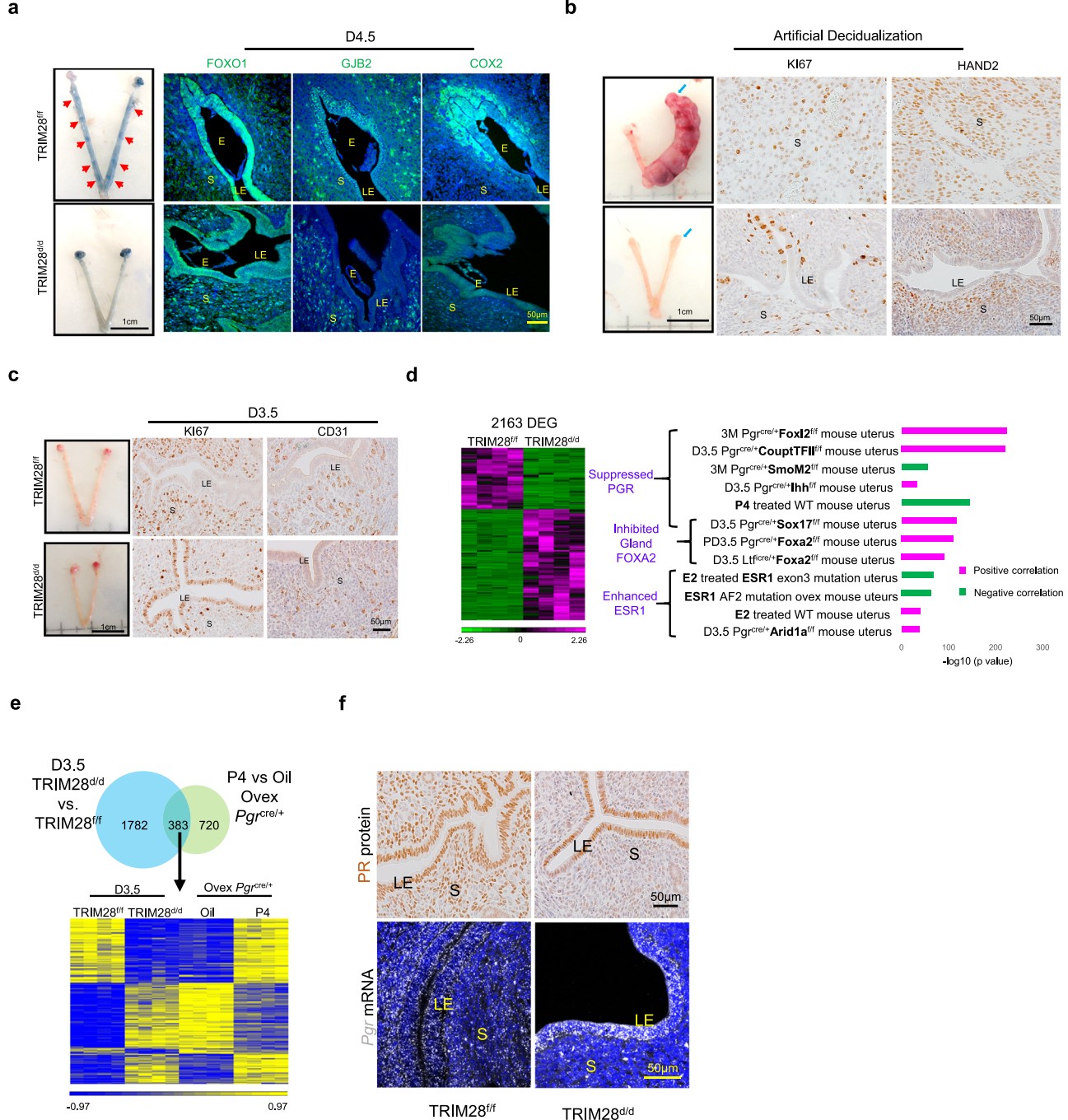

**Fig. 3 | TRIM28 deletion impaired decidualization and steroid signaling in mouse uterus. a** The representative pictures and immunofluorescence of FOXO1, GJB2, and COX2 at D4.5 uterus. Red arrows pointed to the embryo implantation sites. **b** The representative uterine pictures and Immunohistochemistry (IHC) of KI67 and HAND2 of decidual uterus upon artificial decidualization. Blue arrow pointed to the decidual horn. **c** The Immunohistochemistry of proliferative marker KI67 and endothelial marker CD31 at D3.5. **d** Heatmap and top altered pathways in TRIM28[d/d] mice at D3.5 based on RNA-seq data. **e** The comparison of transcriptome from D3.5 TRIM28[d/d] mice and P4 (progesterone) treated ovariectomized *Pgr*[cre/+] mice. **f** Immunohistochemistry of PR and in situ hybridization of *Pgr*. E Embryo, LE Luminal epithelium, S Stroma. IHC was repeat in three different mice per group with similar results.

TRIM28 gene signature was similar to the epithelial transcriptome from the epithelial specific PR knockout mice (*Ltf*[Δcre]*Pgr*[f/f] mice)[36] (Supplementary Fig. 4e) and opposite to the epithelial transcriptome from the epithelial specific PR-B isoform overexpression (*Wnt7a*[cre]*PgrB*[LsL/+])[23] (Supplementary Fig. 4f).

Next we identified the regulation of these epithelial genes by both TRIM28 and PR at a genome-wide scale. 96% (10768/11192) of TRIM28 binding peaks were overlapped with at least one PR peak in the mouse uterus (Fig. 4a, Supplementary data 4). 95% (10242/10768)

of PR-TRIM28 co-occupied regions were also located in previously published[37] uterine chromatin loops. Therefore, we predicted the PR-TRIM28 co-regulated genes based on the overlapping of not only PR-TRIM28 peaks but also the anchor of the associated chromatin loops with the gene regulatory domain. In total, 1175 DEGs from the TRIM28[d/d] uterus can be associated with the PR-TRIM28 peaks. Among them, 545 DEGs showed similar expression patterns with epithelial PR knockout mice and/or opposite expression patterns with the epithelial PR-B overexpression mice (Fig. 4b). These genes

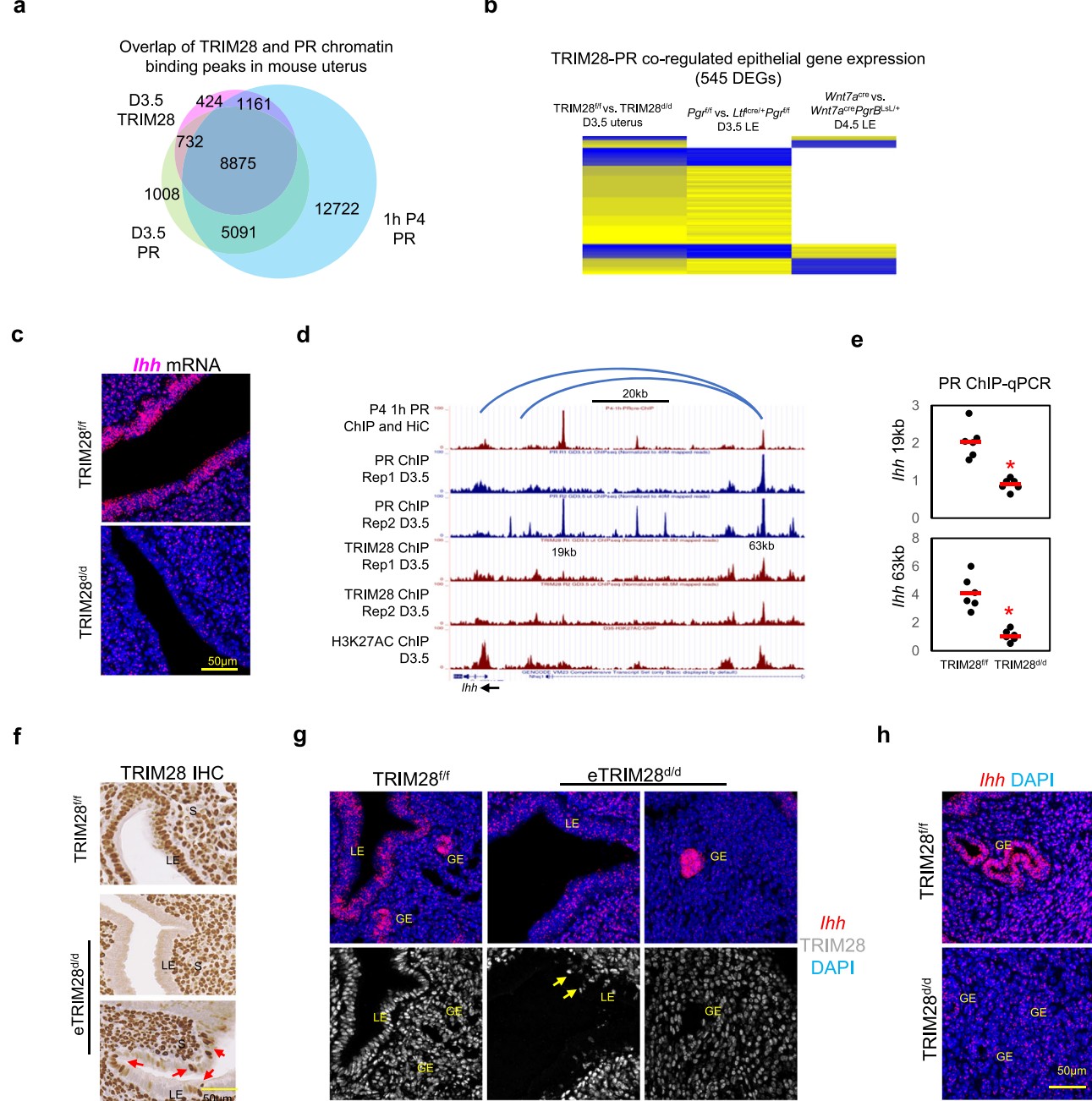

**Fig. 4 | TRIM28 deletion suppressed PR signaling at D3.5 uterine epithelium.**
**a** The overlap of TRIM28 and PR ChIP-Seq in the D3.5 mouse uterus, PR in 1 h P4 (progesterone) treated ovariectomized mouse uterus. **b** The expression heatmap of PR and TRIM28 co-regulated genes. **c** The in situ hybridization (ISH) of *Ihh*. **d** The genome browser of PR, TRIM28 and H3K27AC binding peaks surrounding *Ihh*. Blue line is the Chromatin loops predicted by HiC. **e** The ChIP-qPCR of PR at the enhancers 19 kb and 63 kb upstream of *Ihh* in the TRIM28[f/f] and TRIM28[d/d] mice.

*N* = 6 biologically independent samples per group. Two-sided student's t test.
\**p* < 0.05. **f** Immunohistochemistry (IHC) of TRIM28 in the D3.5 TRIM28[f/f] and eTRIM28[d/d]. **g** In situ hybridization of *Ihh* and immunofluorescence of TRIM28 in the D3.5 TRIM28[f/f] and eTRIM28[d/d]. **h** In situ **h**ybridization of *Ihh* in the D3.5 TRIM28[f/f] and TRIM28[d/d]. GE Glandular epithelium. LE luminal epithelium. Rep replicate. ISH and ICH were repeated in three different mice per group with similar results.

are likely the epithelial genes under direct control of both TRIM28 and PR.

We hypothesized that TRIM28 ablation may alter the PR chromatin binding activity to regulate expression of epithelial PR target genes. Taking the example of *Indian hedgehog* (*Ihh*), an epithelial specific PR target gene that plays crucial roles during embryo implantation[33], *Ihh* expression was reduced by both TRIM28 deletion and epithelial PR knockout but increased by epithelial PR over-expression in the mouse uterus (Fig. 4c, supplementary data 3). There are two PR-TRIM28 co-binding peaks located 19 kb and 63 kb upstream

of the *Ihh* promoter (Fig. 4d). Both 19 kb and 63 kb regions could be brought to the proximity of the Ihh promoter via chromatin loops (Fig. 4d). The 19 kb region is a known enhancer that mediates progesterone-dependent uterine *Ihh* expression[30]. A reduction of PR binding events was found at both the 19 kb and 63 kb upstream of *Ihh* in the TRIM28[d/d] uteri (Fig. 4e). Our finding supports TRIM28 as a modulator of PR occupancy at the 19 kb and 63 kb enhancers to promote *Ihh* transcription. Additionally, we found the PR binding activity has also been reduced in the TRIM28[d/d] uteri at multiple but not all the chromatin loci that are close to other PR-TRIM28 co-regulated genes

(Supplementary Fig. 5) suggesting a gene specific regulatory role of TRIM28 on PR transcription activity.

## Epithelial TRIM28 regulates *Ihh* in a cell specific fashion

We showed that TRIM28 and PR can co-bind at certain chromatin sites to modulate PR transduction activity in the epithelial compartment. However, since $Pgr^{cre/+}$ deletes TRIM28 in both the stromal and epithelial compartments and crosstalk of uterine epithelium and stroma are essential in the uterus[6], it was still unclear whether the suppressed epithelial PR activity was a direct consequence of epithelial TRIM28 deletion. To address this issue, we deleted TRIM28 specifically from the uterine epithelium using $Ltf^{cre/+38}$ to generate the $Ltf^{cre/+}Trim28^{f/f}$ (eTRIM28$^{d/d}$) mice. An incomplete deletion of TRIM28 from the uterine epithelium was found in the eTRIM28$^{d/d}$ mice which showed severe subfertility (Fig. 4f, Supplementary Fig. 6a). No detected embryo implantation at D4.5 uterus with the blastocysts in the uterus and failed uterine responses during artificial decidualization suggested epithelial TRIM28 is also critical for decidualization (Supplementary Fig. 6b–d). Similar to the TRIM28$^{d/d}$ mice, we observed decreased expression of gland specific genes, while enhanced estrogen target genes but suppressed progesterone target genes in the eTRIM28$^{d/d}$ uterine epithelium (Supplementary Fig. 6e), that may lead to the implantation problems.

*Ihh* mRNA was also reduced in the whole uterus of eTRIM28$^{d/d}$ mice (Supplementary Fig. 6e). Double staining of *Ihh* mRNA and TRIM28 protein further indicated *Ihh* expression was decreased in both the TRIM28 negative and positive uterine luminal epithelium (LE) of eTRIM28$^{d/d}$ mice, but remained high in the TRIM28 negative uterine glandular epithelium (GE) of the eTRIM28$^{d/d}$ mice (Fig. 4g). Since *Ihh* expression was also reduced in the uterine glands of the TRIM28$^{d/d}$ mice (Fig. 4h), these results suggested that epithelial TRIM28 was critical for *Ihh* expression in the uterine LE but dispensable for *Ihh* in the GE. Furthermore, we found PR was still highly expressed in the luminal epithelium in the eTRIM28$^{d/d}$ uterus (Supplementary Fig. 6f) supporting that epithelial TRIM28 deletion alone may suppress *Ihh* expression but not affect PR expression. Interestingly, eTRIM28$^{d/d}$ mice did not show reduced stromal cell proliferation or decreased PR and PR/ERα target gene expression at the stroma which have been observed in the TRIM28$^{d/d}$ mice and are essential for decidualization[16,39] (Supplementary Fig. 6f, g) suggesting that stromal and epithelial TRIM28 play critical and distinct roles in uterine biology.

## TRIM28 deletion disrupted ERα induced stromal *Pgr* expression

TRIM28 ablation in *Pgr*-expressing cells resulted in a reduction in PR levels in mouse uterine stroma (Fig. 3f). Since stromal *Pgr* is positively regulated by the estrogen signaling through stromal ERα[40], we examined the impact of TRIM28 loss on uterine estrogen signaling. In general, estrogen signaling was enhanced but hundreds of estrogen target genes remained suppressed in the TRIM28$^{d/d}$ uterus, such as *Igf1*, *Egr1*, *Fst*, and *Has1* despite of an unaltered ERα protein level (Fig. 5a, Supplementary Fig. 7a–c).

At the genome-wide level, TRIM28 and ERα occupy 6745 common genomic regions and share 882 common downstream target genes in the uterus (Fig. 5b, Supplementary data 4). 462 common target genes have both TRIM28 and ERα occupancy within the defined proximity and were involved in well-known estrogen related pathways, such as BMP, cholesterol synthesis, and IGF1 (Fig. 5b). These findings implied that TRIM28 may also modulate the ERα transduction.

Based on TRIM28's role in the modulation of PR genome occupancy, we hypothesized that TRIM28 is also required to permit ERα occupancy at the *Pgr* locus for transcriptional regulation. Multiple ERα occupying regions are found surrounding the *Pgr* gene (Fig. 5c). Among them, six had TRIM28 co-occupancy, H3K27AC positive enhancer labeling and were linked to the *Pgr* promoter by chromatin loops (Fig. 5d). Assessing ERα binding events by ChIP-qPCR revealed a reduction of ERα occupancy at the *Pgr* promoter and the 3′ UTR (Fig. 5d).

## TRIM28 interacts with PR and ERα in human and mouse endometrium

Similar to the decidual HESCs, we found TRIM28, ERα and PR can form a complex in the D3.5 mouse uterus (Fig. 6a) indicating the conserved co-regulation of these proteins in both human and mouse. Indeed, we observed a significant overlap of TRIM28, ERα and PR chromatin binding in the mouse uterus (Fig. 6b, Supplementary Fig. 7c), and co-regulated 918 DEGs (Supplementary Fig. 7d).

Since the DEGs from D3.5 TRIM28$^{d/d}$ uterus reflected the combined effects of TRIM28 on ERα and PR signaling, we further validated the functions of TRIM28 on ERα or PR respectively using the E2 or P4 treated ovariectomized mouse models. We investigated the well-known ERα and PR target genes using qPCR. To our surprise, all the E2 induced genes, including genes that were increased in the D3.5 TRIM28$^{d/d}$ uterus, were suppressed in the E2 alone treated TRIM28$^{d/d}$ uterus, indicating that ERα signaling was mainly suppressed by TRIM28 deletion (Fig. 6c). The PR target genes were still reduced in the P4 alone treated TRIM28$^{d/d}$ uterus validating TRIM28 deletion suppressed PR signaling (Fig. 6c). Therefore, the enhanced estrogen target genes in the D3.5 TRIM28$^{d/d}$ uterus most likely reflect the combined effects of disrupted PR and ERα signaling.

Similarly, the TRIM28 ChIP-Seq from the pre-decidual HESCs and the published PR and ERα ChIP-Seq from the human endometrium[5,41] also exhibited a large number of overlapped peaks, suggesting that TRIM28 may also play an important role in mediating the ERα and PR signaling in human endometrium (Fig. 6d, Supplementary Fig. 7e). To directly unveil the functional coupling between TRIM28 and PR/ERα, we performed luciferase reporter assay in Human endometrial cancer cell line (HEC1A). We found TRIM28 can further enhance two PR isoforms, PRA/PRB and ERα transcription activity at the 19 kb upstream enhancer of *Ihh* (Fig. 6e, Supplementary Fig. 7e, f), indicating that TRIM28 directly facilitate the transcriptional regulatory functions of PR/ERα.

## TRIM28 deletion alters the cellular composition of the mouse uterine stroma

Since *Pgr* started to express in the majority of uterine cells after puberty, $Pgr^{cre/+}$ mediated *Trim28* ablation occurred mainly in the uterine epithelium at postnatal day (PND) 14 and in most uterine stromal and epithelial cells by PND21 (Supplementary Fig. 8a). At 8-week-old TRIM28 was deleted in the TRIM28$^{d/d}$ uterine stromal cells which had low but detectable PR expressions (Fig. 7a and Supplementary Fig. 8a). Surprisingly, TRIM28 immunopositivity was still preserved in another subset of stromal cells in which PR protein was not detected (Fig. 7a and Supplementary Fig. 8a). To rule out the possibility that these TRIM28 positive cells were transformed from the *Pgr* lineage, we crossed the TRIM28$^{d/d}$ mice with the Sun1-GFP reporter mice that produces peri-nuclear GFP staining in cre expressing cell lineages[42]. In the TRIM28$^{d/d}$ mice, no GFP expressing cells showed TRIM28 staining (Fig. 7a). This indicated an efficient ablation of TRIM28 in PR-expressing cells but also suggested the expansion of the PR-negative (−) and TRIM28-positive (+) stromal cells in the TRIM28$^{d/d}$ endometrium.

Three D3.5 TRIM28$^{f/f}$ uteri were pooled together then four D3.5 TRIM28$^{d/d}$ uteri were pooled together for scRNA-Seq to determine the cell characteristics of these PR-/TRIM28+ stromal cells in the TRIM28$^{d/d}$ uteri. We annotated the identity of 23 cell clusters according to the known marker genes (Fig. 7b, Supplementary data 5). Six clusters were linked to the fibroblast (Fibr) as defined by common mesenchymal markers *Vim* and *Pdgfrb*. Three clusters were pericytes (Peri) with common pericyte markers *Acta2* and *Mcam*. Among them, *Ctla2a*, *Hsd11b2* Fibr and two proliferative *Pcna* and *Mki67* Fibr were mainly

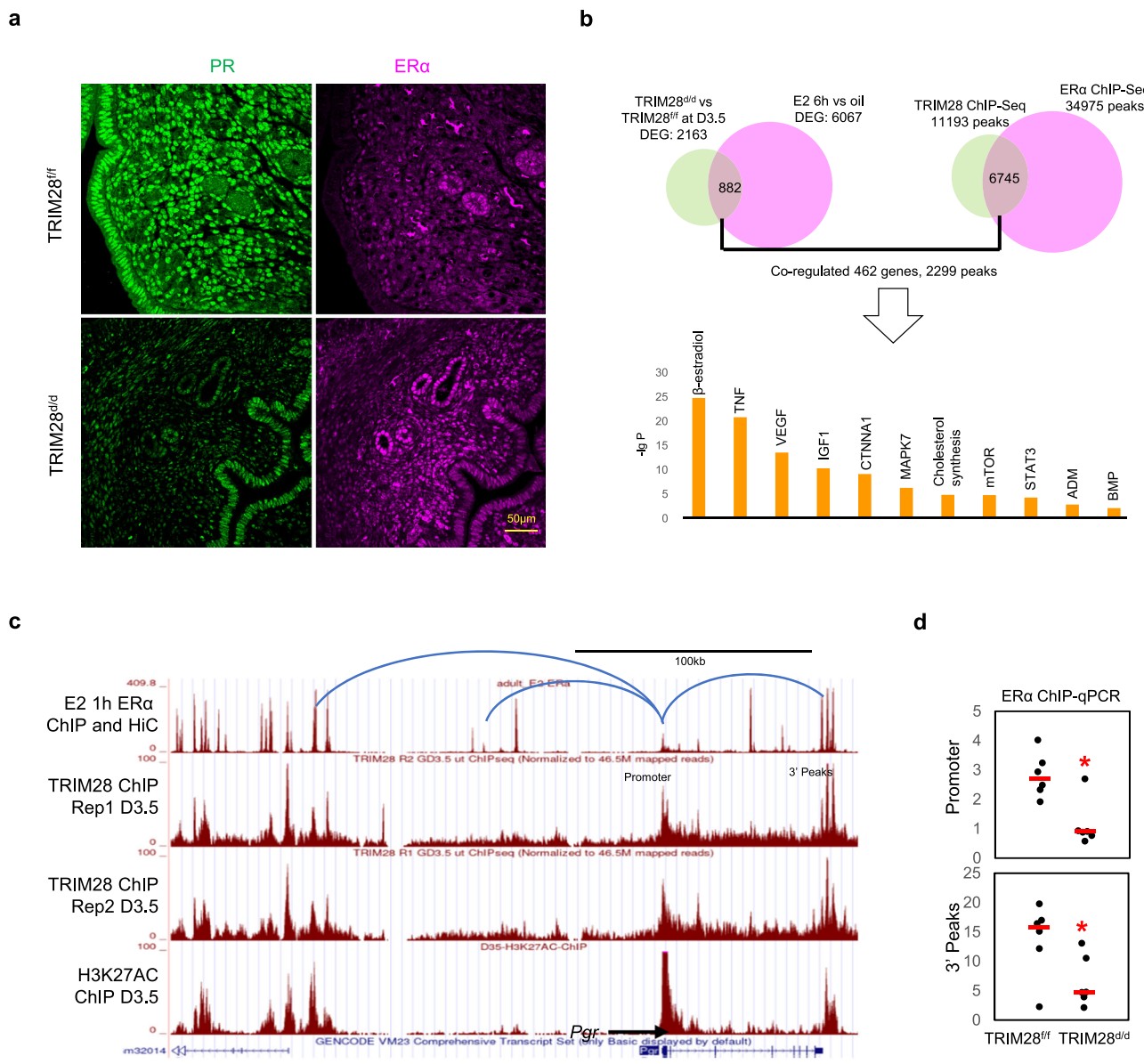

**Fig. 5 | TRIM28 deletion inhibited ERα activity. a** Immunofluorescence (IF) of PR and ERα. **b** The overlapped binding peaks of TRIM28 and ERα, the associated common DEGs and top altered pathways between TRIM28^d/d^ mice and E2 treated ovariectomized wildtype mouse uterus. **c** The genome browser of ERα, TRIM28 and H3K27AC binding peaks at *Pgr*. Blue line is the Chromatin loops predicted by HiC.

**d** The ChIP-qPCR of ERα binding activity at the promoter and 3′ peak enhancer. Two-sided student's t test. *$p < 0.05$. $N = 6$ biologically independent samples per group. Rep: replicate. The IF was repeated in three different mice per group with similar results.

found in the TRIM28^f/f^ uterus, while *A2m* and *Lars2* Fibr were the dominant in the TRIM28^d/d^ uterus. *Rgs5* Peri was the major pericyte in the TRIM28^f/f^ uterus, but *Pcp4* and *Mustn1* Peri were found more in the TRIM28^d/d^ uterus.

We posited the alteration of the cell composition in the TRIM28^d/d^ uterus may lead to the accumulation of TRIM28+ cells. Therefore, we calculated the average expression of *Trim28* and *Pgr* in each cell cluster. According to the immunofluorescence results, TRIM28 was found in almost all the TRIM28^f/f^ stromal cells, so any cell clusters with *Trim28* expression that was comparable to the TRIM28^f/f^ stromal cells were defined as the *Trim28* + . *Pgr* was not detected in the endothelial cells of the TRIM28^f/f^ uterus. Therefore, any cell clusters with higher *Pgr* RNA abundance than the endothelial clusters were defined as *Pgr* + . Results from all the non-epithelial cell clusters were shown as the pie charts (Fig. 7c). We found the majority of fibroblasts in the TRIM28^d/d^ uterus were *Trim28*-, while the *Mustn1* Peri, Endo and all the immune clusters were most likely *Trim28+/Pgr−*. Additionally, we found *Pcp4*

Peri were *Pgr + /Trim*28+ which may explain the identity of some PR + / TRIM28+ cells in the myometrial layers of the TRIM28^d/d^ uterus (Supplementary Fig. 8a). In summary, we identified the *Trim28 + /Pgr−* cells in the TRIM28^d/d^ stroma were most likely attributed to the *Mustn1* pericytes and immune cells.

Since the TRIM28 was efficiently deleted from the uterine fibroblasts in the TRIM28^d/d^ uterus, we next compared the TRIM28^d/d^ *A2m* Fibr with the TRIM28^f/f^ *Hsd11b2* Fibr to determine the in vivo effects of TRIM28 deletion in fibroblasts (Fig. 7d). Similar to the whole uterus, inflammation IFNG, estrogen, and PR antagonist mifepristone mediated pathways were enriched in the TRIM28^d/d^ fibroblasts. The proliferative signaling Myc was suppressed in the TRIM28^d/d^ fibroblasts which is consistent with the reduced stromal proliferation by immunohistochemistry and our in vitro study of si*TRIM28* treated HESCs. Additionally, TRIM28 deletion may regulate histone demethylation KDM1A, protein translation EIF2 or post-translational modification OGT in the fibroblasts. These results indicate that TRIM28 deletion

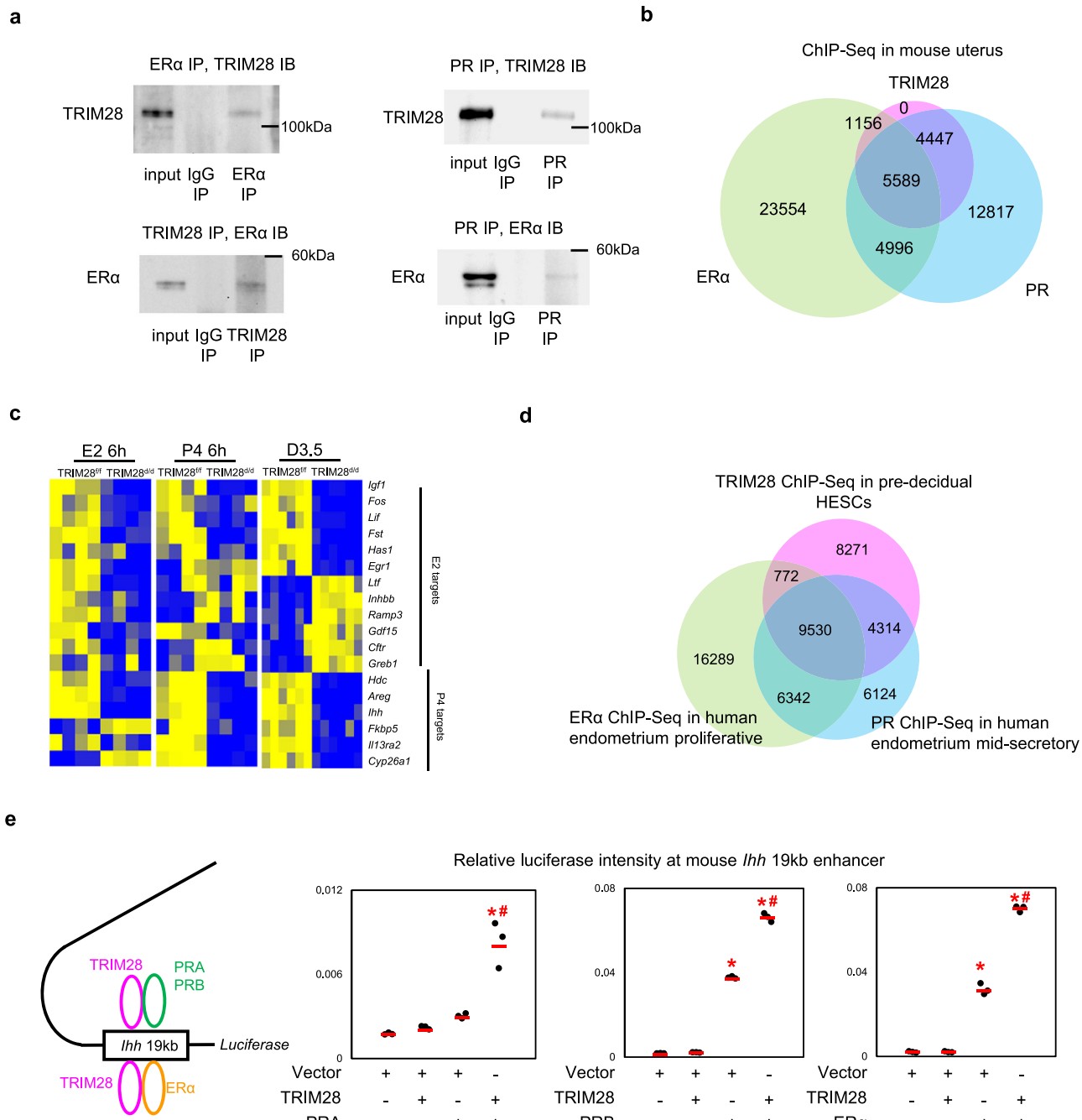

**Fig. 6 | ERα, PR and TRIM28 interacts with each other in the mouse uterus.**
**a** The co-immunoprecipitation (co-IP) of PR, TRIM28, ERα in mouse uterus. IP: immunoprecipitation. IB: Immunoblot. $N = 3$ mice. **b** The overlap of TRIM28 ChIP-Seq at D3.5 mouse uterus, PR ChIP-Seq in 1 h Progesterone (P4) treated uterus and ERα ChIP-Seq in 1 h 17β-estradiol (E2) treated uterus. $N = 4$ mice. **c** The heatmap of fold changes of E2 and P4 target genes in the P4 6 h or E2 6 h treated ovariectomized TRIM28^d/d uterus compared to vehicle, or D3.5 TRIM28^d/d uterus compared to the TRIM28^f/f uterus. **d** The overlap of TRIM28 ChIP-Seq in pre-decidual

HESCs, and published PR ChIP-Seq in human endometrium at mid-secretory phase, ERα ChIP-Seq in human endometrium at proliferative phase. **e** Relative luciferase intensity at mouse *Ihh* 19 kb enhancer in HEC1A cells co-transfected with TRIM28 and PRA or PRB or ERα. $N = 150,000$ cells examined over three independent experiments. One-way ANOVA with post-hoc turkey's test. *$p < 0.05$, compared to vector. #$p < 0.05$, compared to PRA or PRB or ERα alone. The co-IP was repeated in three different mice with similar results.

transforms uterine fibroblasts toward a pro-fibrosis and anti-proliferative cellular phenotype which may explain the reduction of *Pgr* lineage cells from the TRIM28^d/d stroma.

### The characteristics of TRIM28 positive PR negative cells in the uterine stroma

As noted above, *Mustn1* pericytes were a subset of expanded *Trim28*+ / *Pgr*− cells in the TRIM28^d/d uterus. Functional annotation of the marker

genes indicated *Mustn1* pericytes were enriched with the classical myocyte signaling including cardia hypertrophy, fibrosis, and calcium, and, more importantly, the cell pluripotency pathway (Fig. 7e). Since the peri-vascular pericytes have been identified as mesenchymal progenitor/stem cells in the human endometrium[43], we also checked some cell differentiation marker *Mustn1*, *Aspn* and mesenchymal stem marker *Pdgfrb*, *Mcam*, which were also enriched in this pericyte cluster (Supplementary Fig. 8b).

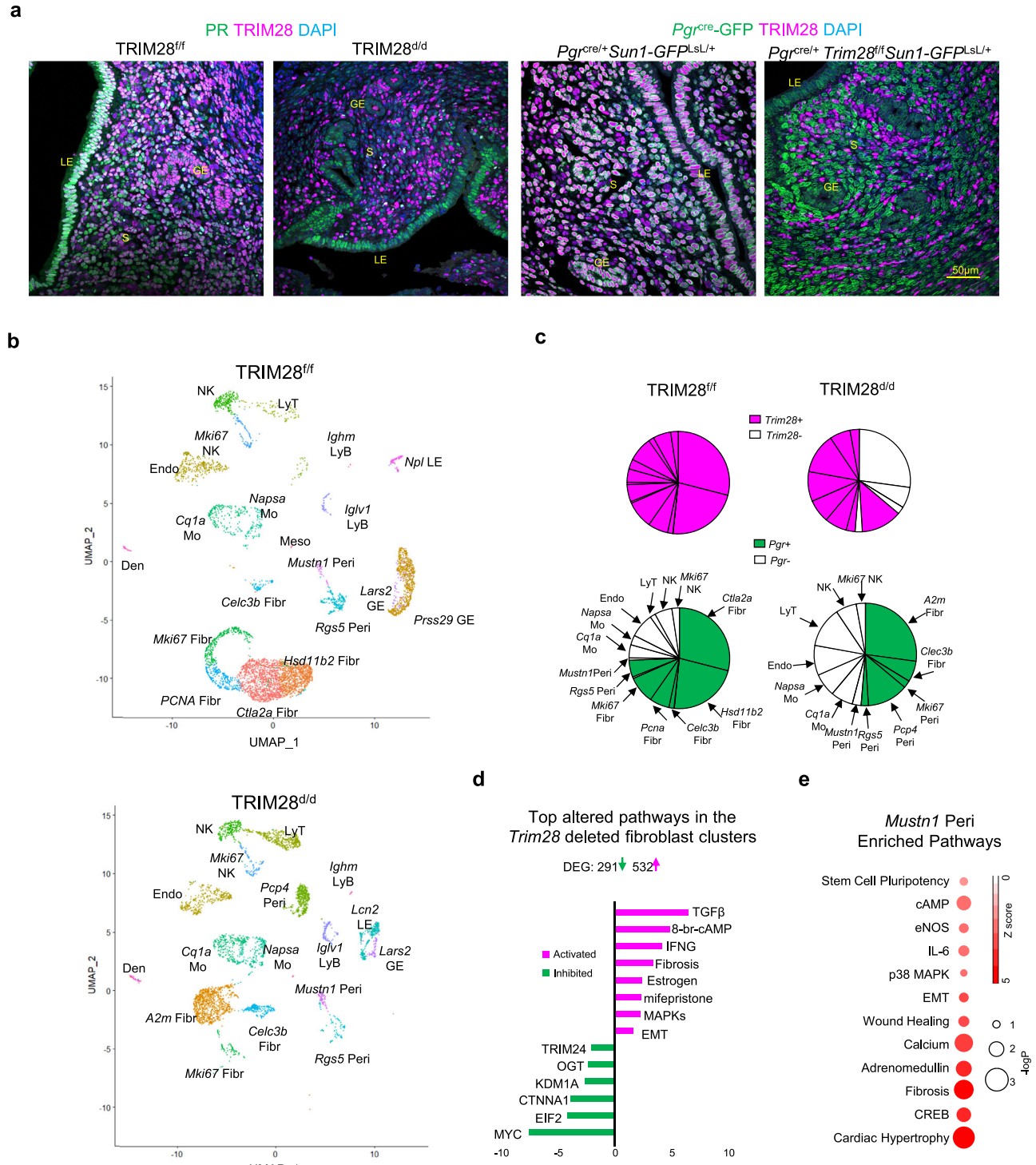

**Fig. 7 | Abnormally accumulation of TRIM28 positive alone cells in the TRIM28d/d uterus. a** Immunofluorescence (IF) of TRIM28 and PR in the TRIM28$^{f/f}$ and TRIM28$^{d/d}$ mice at D3.5. Immunofluorescence of TRIM28 and *Pgr*$^{cre}$-GFP in the *Pgr*$^{cre/+}$*Sun1-GFP*$^{LsL/+}$ and *Pgr*$^{cre/+}$*Trim28*$^{f/f}$ *Sun1-GFP*$^{LsL/+}$ mice at D3.5. **b** UMAP of scRNA-Seq in the TRIM28$^{f/f}$ and TRIM28$^{d/d}$ mice at D3.5. **c** The pie plot of all the non-epithelial clusters. Each sector represented one cell cluster. The area of the sector is proportional to the number of the cells in each cluster. The TRIM28+ is magenta. The PGR+ is green. **d** The top altered pathways in the *A2m* Fibr from the TRIM28$^{d/d}$

mice compared with the *Hsd11b2* Fibr from the TRIM28$^{f/f}$ mice. Green means inhibition. Magenta means activation. **e** Enriched pathway in *Mustn1* Peri annotated by IPA based on the cluster specific marker genes. The size of the red dot is proportional to the -logP value. The red color is proportional to the z score. Fibr Fibroblast, Peri Pericytes, LE Luminal epithelium, GE Glandular epithelium, Endo Endothelium, Mo Myeloid, NK Natural killer cells, LyT Lymphocyte T, LyB Lymphocyte, Den Dendritics, Meso Mesothelium. The IF was repeated in three different mice per group with similar results.

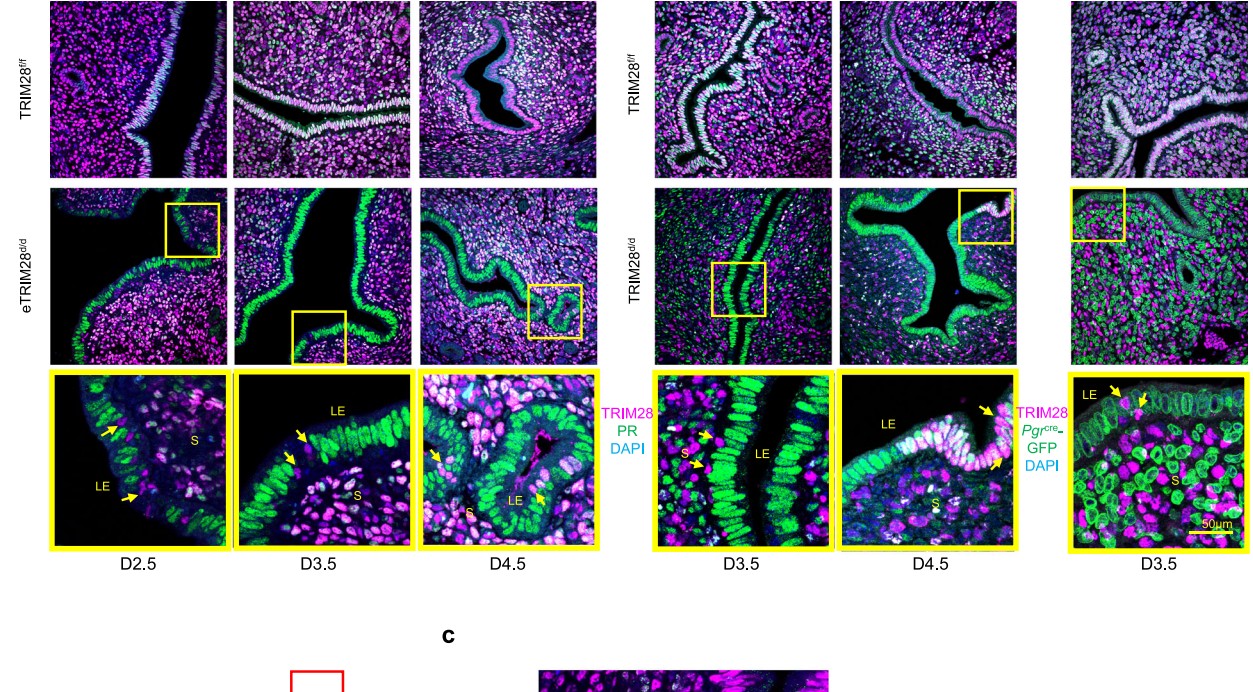

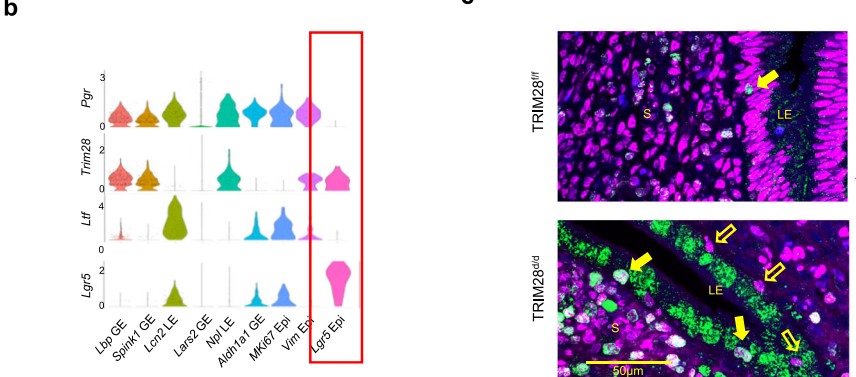

**Fig. 8 | Epithelial TRIM28 deletion reduced Ihh expression and promoted progenitor cell accumulations at luminal epithelium. a** The immuno-fluorescence (IF) of TRIM28 and PR, TRIM28 and *Pgr*^cre-GFP at D2.5, D3.5, and D4.5 of the TRIM28^f/f, eTRIM28^d/d and TRIM28^d/d mouse uterus. Arrow points to the TRIM28 positive cells in the luminal epithelium of the TRIM28^d/d and eTRIM28^d/d mice. **b** The violin plot of *Pgr*, *Trim28*, *Esr1*, *Lgr5* single cell expressions. **c** The immunofluorescence of TRIM28 and LGR5. Closed arrow refers to TRIM28 and LGR5 positive epithelium. Open arrow refers to TRIM28 positive but LGR5 negative epithelium. LE Luminal epithelium, GE Glandular epithelium, Epi Epithelium. was repeated in three different mice per group with similar results.

We re-clustered all the cells that expressed the mesenchymal makers *Pdgfra*, *Pdgfrb*, *Acta2*, or *Vim*, including the fibroblast, pericyte and myeloid clusters (Supplementary Fig. 8c) and inferred cell trajectory using Monocle[44] and RNA velocity[45]. As expected, cell transitions within the fibroblast clusters have been observed in the TRIM28^d/d (from *A2m* Fibr to *Clec3b* Fibr) and TRIM28^f/f (*from Hsd11b2*, *Ctla2a* Fibr toward *Pcna*, *Mki67* Fibr) fibroblasts and pericytes (between *Mustn1* and *Rgs5* Peri) (Supplementary Fig. 8d). And a transition between the pericytes and fibroblasts have been predicted only by Monocle (Supplementary Fig. 8d) which triggers an interesting cell development proposal but requires further experiments to support.

**TRIM28 positive PR negative progenitor cells in the uterine epithelial compartment**

Similar to the accumulated TRIM28+ cells in the TRIM28^d/d stroma, TRIM28 + /PR- was detected in the eTRIM28^d/d and TRIM28^d/d uterine LE at D2.5 and D3.5 (Fig. 8a). However, PR staining was observed in some of the TRIM28+ cells at D4.5 (Fig. 8a), implying the transformation of these TRIM28+ cells from PR- toward PR+ in the uterine LE. Furthermore, in the D3.5 *Pgr*^cre/+*Trim28*^f/f*Sun1-GFP*^LsL/+ mice, we also

detected TRIM28 + /GFP- uterine LE, indicating non-*Pgr* lineage identity (Fig. 8a).

In order to understand the cell identity of the TRIM28+ epithelium, we reviewed our D3.5 scRNA-Seq data. UMAP analysis indicated that the TRIM28^f/f and TRIM28^d/d mice had distinct transcriptome in both GE and LE including the suppressed progesterone and enhanced estrogen signaling but enhanced EMT and/or pluripotency pathways in the TRIM28^d/d GE and LE, suggesting an enhanced progenitor activity (Supplementary Fig. 9a). Re-clustering of all the epithelial clusters based on the epithelial marker expression *Krt8*, *Epcam* and *Cdh1* revealed more cell subtypes at the uterine epithelium (Supplementary Fig. 9b). Among them, the *Lgr5* Epi was the only epithelial cluster that showed low expressions of *Pgr* but high expression of *Trim28* (Fig. 8b), which could be the source of TRIM28 + /PR- epithelium. Additionally, the stem cell marker *Lgr5*[46] was also expressed at a significantly higher level in several TRIM28^d/d epithelial cluster including *Lcn2* LE, *Aldh1a1* GE and *Mki67* Epi compared to the rest of TRIM28^f/f clusters.

The single cell trajectory inference showed a possible cell development route from *Lars2* GE to *Prss29* GE, and a bifurcation from the *Lgr5* Epi toward *Npl* LE and the *Lars2* GE in the TRIM28^f/f epithelium

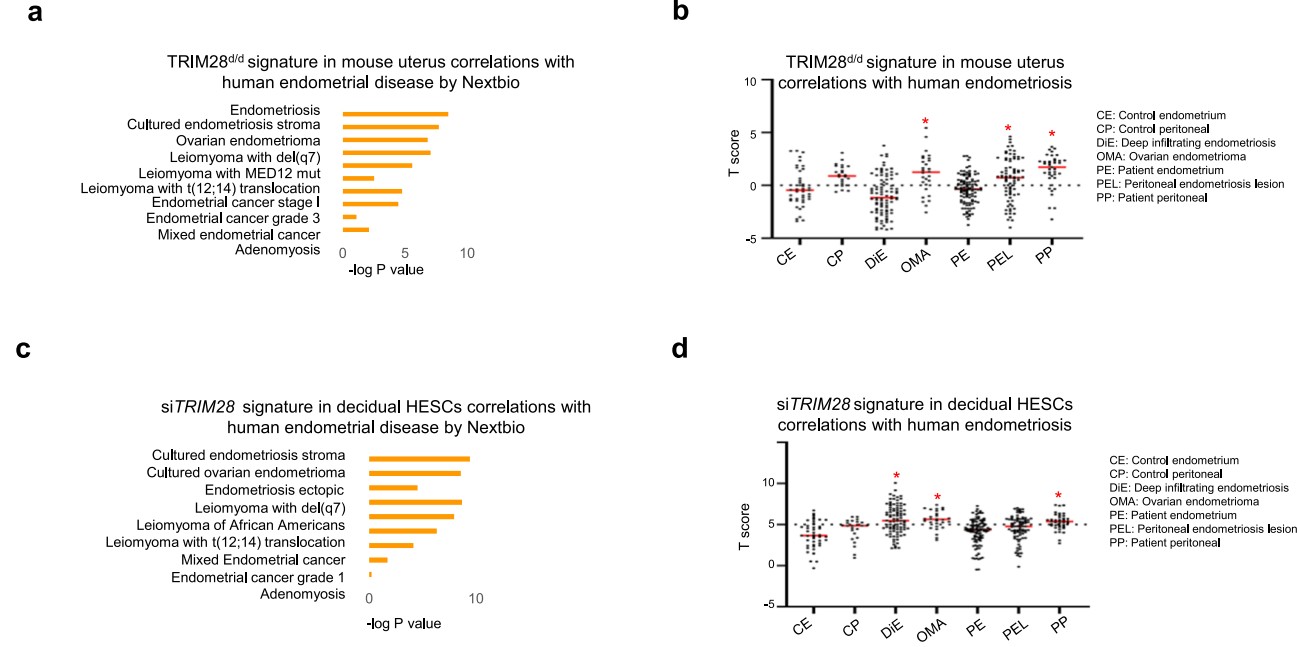

**Fig. 9 | TRIM28 signature correlated with human endometrial diseases.** The correlation *p* value **a**, **c** and the T score of gene signature **b**, **d** from the D3.5 TRIM28^d/d uterus **a**, **b** and the si*TRIM28* treated decidual HESCs **c**, **d** compared with the multiple human endometrial diseases **a**, **c** and human endometriosis transcriptome from different subtypes **b**, **d**. $N = 408$ biologically independent samples in total from human endometriosis datasets for **b** and **d**. one-way ANOVA with post-hoc turkey's test was used. *$p < 0.05$ compared to CE.

(Supplementary Fig. 9c). It posits a possibility that *Lgr5* Epi may be the bipotent stem cells for LE and GE reported before[47]. A completely different cell trajectory was predicted for the TRIM28^d/d epithelium which may start from the *Aldh1a1* GE toward the *Lars2* GE, or toward the *Lcn2* LE and *Mki67* Epi (Supplementary Fig. 9d). LGR5 protein was highly expressed in the majority of cells within the TRIM28^d/d epithelium while only in very few TRIM28^f/f epithelia (Fig. 8c, Supplementary Fig. 9e). In combination of the increased stem cell marker LGR5 expression and enhanced cell pluripotency pathway in the TRIM28^d/d epithelium, the TRIM28^d/d epithelium seems promote the presence of progenitor cells at the expense of more differentiated cells.

### The TRIM28 gene signature positively correlates with human endometriosis

Collectively, our data show that TRIM28 deficiency results in uterine progesterone resistance and estrogen hypersensitivity. These aberrant hormone responses are also seen in numerous uterine disorders[2,3]. Indeed, the transcriptomic profile of TRIM28^d/d uteri resembles multiple human endometriosis, leiomyoma and endometrial cancer datasets[48–54] (Fig. 9a). Because human endometriosis has the highest resemblance with the TRIM28 dependent transcriptome, we further examined TRIM28's association with various types of endometriosis from the EndometDB, an endometriosis dataset with multiple subtypes of endometriosis[55]. Compared to control endometrial tissues, mouse TRIM28^d/d gene signature was positively correlated with Ovarian endometrioma (OMA) and Patient peritoneal (PP).

Based on the progesterone resistance phenotype that we observed in the TRIM28 deficient HESCs under the hormone stimulation, it was not surprising that TRIM28 gene signature in the decidual HESCs also showed high resemblance with multiple human endometriosis datasets[55–57], especially a positive correlation with multiple endometriosis subtypes including the Deep infiltrating endometriosis (DiE), OMA, and PP in comparison with the control endometrium (Fig. 9c, d).

In summary, we identified a positive correlation between the TRIM28 deletion induced gene signature with human endometrial disorders, such as endometriosis, underscoring a crucial role for TRIM28 in these endometrial pathologies.

## Discussion

Using cultured HESCs and genetically engineered mouse models, we identified the essential functions of TRIM28 in modulating estrogen and progesterone signaling in human and mouse endometrial cells. Furthermore, TRIM28 deletion not only altered the characteristics of PR positive endometrial stromal cells, but also promoted the enrichment of PR negative and TRIM28 positive pericytes and epithelium with progenitor potential in the mouse uterus.

We demonstrated that TRIM28 can modulate ERα and PR activity in the human and mouse endometrium through altered chromatin binding activity. TRIM28 has been widely recognized as a transcriptional co-repressor[58]. In our study, we proposed that TRIM28 acted as a transcriptional activator of ERα and PR. From our RNA-Seq analysis, we found TRIM28 deletion can suppress the histone demethylation KDM1A and O-GlcNAc transferase OGT pathway, which may affect PR and ERα activity as reported before[59,60], disrupt Sumo- and Ubiquitin- E3 ligase[61,62], which may enhance ERα and PR activity through sumo- and ubiquitin- modification[63–65]. Additionally, the crosstalk of TRIM28 with STAT, NOTCH, SMAD proteins that we found in the pre-decidual HESC have already been proposed in the vasculature, immune cells, and cancer cell lines[66–68], indicating critical roles of TRIM28 in transducing the crosstalk of signaling. Since TRIM28 contains large intrinsically disordered regions and can facilitate chromatin compartmentalization[69], we hypothesized that TRIM28 may be a fundamental molecule of the transcriptional condensate[70] that play a central role in cell biology. Further study is required to unveil the formation of transcriptional condensates in the uterus and how TRIM28 may be involved.

In the control mouse uterus, the majority of stromal and epithelial cells was derived from the PR expressed cell lineage. However, our study in TRIM28^d/d uterus revealed a subset of PR-TRIM28+Mustn1+ pericytes and the PR-TRIM28 + LGR5+ epithelial cell type as the progenitors of the stromal and epithelial cells, respectively. Currently, there is no consensus concerning the marker genes and steroid

receptor expression status of the uterine stem cells[71,72]. Both the tissue resident and bone marrow derived endometrial stem/progenitor cells are involved in the regeneration of uterus at both physiological and pathological processes[71]. The fact that TRIM28 is highly expressed in the human stem cells to maintain the pluripotency[10] may explain its essential role in different types of progenitor cells. Since PR-TRIM28+ progenitor cells were highly enriched when the hormone response of normal uterine cells was disrupted, they may play more dominant roles in hormone disrupted pathology observed in many endometrial diseases.

We found that TRIM28 deletion induced a gene signature in the mouse uterus and HESCs that positively correlated with several endometrial disorders, especially endometriosis[73]. The estrogen hypersensitivity and progesterone resistance are regarded as pivotal factors for the causation and promotion of this gynecological disorder[74]. Hormone therapy is the main non-surgical therapy for treating endometriosis despite of its detrimental effects on fertility and limited efficacy long term[74]. Therefore, we believe that further study of TRIM28 modulation of ER α and PR activity will provide the insight towards developing non-hormonal therapy for endometrial diseases and, more importantly, multiple systematic diseases including the immune disturbances, muscular hypotrophy and obesity that have been linked to TRIM28 deletion[11–13].

## Methods

### Mice
B6.129S2(SJL)-Trim28[tm1.1Ipc]/J, *Trim28*[f/f] mice[75] and Ltf[tm1(icre)Tdku]/J, *Ltf*[tcre/+] mice[38] were purchased from the Jackson Laboratory. *Pgr*[cre/+] mouse colony[20] was maintained in our lab. They were bred to generate the reproductive tract specific TRIM28 knockout *Pgr*[cre/+]*Trim28*[f/f] (TRIM28[d/d]) mice and uterine epithelial specific TRIM28 knockout *Ltf*[tcre/+]*Trim28*[f/f] (eTRIM28[d/d]) mice. All animal studies were conducted in accordance with the Guide for the Care and Use of Laboratory Animals published by the National Institutes of Health and animal protocols approved by the Institutional Animal Care and Use Committee (IACUC) at the National institute of Environmental Health Sciences (NIEHS).

### Cultured cells
Human endometrial samples were obtained from three healthy, reproductive-aged volunteers with regular menstrual cycles and no history of gynecological malignancies under a human subject protocol approved by the institutional review board of Baylor College of Medicine. All donors provided written informed consent. Primary human endometrial stromal cells (HESCs) were cultured in DMEM-F12 supplemented with 5% Fetal bovine serum (FBS) and 1X Penicillin Streptomycin (P/S). The primary cells that before passage 6 were used for the experiments. Hela cells (CCL-2) were purchased from ATCC and cultured in ATCC-formulated Eagle's Minimum Essential Medium.

### Hormone treatment of HESCs
The decidualization of primary HESCs were induced by EPC treatment in which opti-MEM medium supplemented with 2% charcoal stripped FBS and 1× P/S and a hormone mixture (EPC) including 10 nM 17β-Estradiol, 1 μM Medroxyprogesterone acetate, and 100 μM $N^6,2'$-O-Dibutyryladenosine 3′,5′-cyclic monophosphate sodium salt (db-cAMP).

### TRIM28 knock down by small-interfering RNA (siRNA)
The primary HESCs were transfected with 60 nM nontargeting siRNA (si*NT*) or siRNA targeting *TRIM28* (si*TRIM28*, ON-TARGETplus SMARTpool; Dharmacon) using Lipofectamine RNAiMax according to the manufacturer's instructions.

### Edu labeled cell proliferation
After 48 h of si*NT* and si*TRIM28* treatment, the cells were changed to the culture medium added with 10 μM EdU for 24 h. Then the slides

were stained by Click-iT® EdU Imaging Kits following the manual. The proliferative cells that incorporated EdU during 24 h culture period exhibited positive staining of GFP. The intensity of GFP signals were quantified using Metamorph software from three random regions in each treatment group and each primary HESC. The primary HESCs from three donors were used for this analysis.

### Cell migration by transwell
After 48 h of si*NT* and si*TRIM28* treatment, primary HESCs in the serum free culture medium were seeded at the insert of the 12-well transwell. In total 10% FBS supplemented culture medium were added at the bottom well. After 24 h culture, the cells at the upside of the insert were wiped by cotton stick. The insert was fixed with 70% ethanol for 10 min, dried, then stained with 0.05% crystal violet for 30 min. The pictures of the cells at the bottom side of the insert was taken under the bright field microscope. The primary HESCs from three donors were used for this analysis.

### Cell apoptosis
After 48 h of siNT and siTRIM28 treatment, the cells were changed to the culture medium for 24 h. Fixed by 4% paraformaldehyde (PFA) for 10 min at RT, the slides were submitted to the NIEHS histology core for Tunel staining. The fixed cells incubated with DNase I for 1 h at 37 °C were used for positive control. The primary HESCs from three donors were used for this analysis.

### Lentivirus transduction and plasmid transfection
HA tagged human *PGR-B* lentiviral plasmid was kindly provided by Dr. Dean Edward in Baylor College of Medicine. GFP tagged TRIM28 plasmid was purchased from Addgene (Plasmid #65397[76]). The lentivirus plasmid was packaged by the virus core at NIEHS. The Hela cells were transduced with lentivirus *PGR-B* plasmid and change to fresh medium after 24 h. The successfully transfected cells were selected by flow cell sorting using the green fluorescence protein ZsGreen that was embedded in the plasmid vector in the flow cytometry core at NIEHS. The PRB positive cells were further transfected with the GFP-TRIM28 plasmids using Lipofectamine 2000 following the manual's protocols. After transfections, the PRB and TRIM28 positive Hela cells were collected for co-IP analysis.

### Rapid immunoprecipitation mass spectrometry of endogenous proteins (RIME)
Primary HESCs from three different donors were decidualized by EPC treatment for three days. The three primary HESCs were fixed by 1% PFA for 8 min at RT and mixed as one sample and set for RIME service to Active Motif. The PR and IgG RIME analysis (supplementary data 6) were performed twice and IgG is the control. The proteins that were enriched in both two technical replicates were listed as potential PGR interacting proteins ($N = 2$ technical replicates).

For RIME service in active motif, an aliquot of chromatin (100 − 150 ug) was precleared with protein G agarose beads (Invitrogen). Proteins of interest were immunoprecipitated using 10-15 ug of antibody against PR and protein G magnetic beads. Protein complexes were washed then trypsin was used to remove the immunoprecipitate from beads and digested the protein sample. Protein digests were separated from the beads and purified using a C18 spin column (Harvard Apparatus). The peptides were vacuum dried using a speedvac.

Digested peptides were analyzed by LC-MS/MS on a Thermo Scientific Q Exactive Orbitrap Mass spectrometer in conjunction with a Proxeon Easy-nLC II HPLC (Thermo Scientific) and Proxeon nanospray source. The digested peptides were loaded on a 100 micron × 25 mm Magic C18 100 Å 5U reverse phase trap where they were desalted online before being separated using a 75 micron × 150 mm Magic C18 200 Å 3U reverse phase column. Peptides were eluted using a 90 minute gradient with a flow rate of 300 nl/min. An MS survey scan

was obtained for the m/z range 300–1600, MS/MS spectra were acquired using a top 15 method, where the top 15 ions in the MS spectra were subjected to HCD (High Energy Collisional Dissociation). An isolation mass window of 1.6 m/z was for the precursor ion selection, and normalized collision energy of 27% was used for fragmentation. A five second duration was used for the dynamic exclusion.

Tandem mass spectra were extracted and analyzed by PEAKS Studio version 8 built 20. Charge state deconvolution and deisotoping were not performed. Database consisted of the Uniprot database (version 180508, 71,771 curated entries) and the cRAP database of common laboratory contaminants (www.thegpm.org/crap; 114 entries). Database was searched with a fragment ion mass tolerance of 0.02 Da and a parent ion tolerance of 10 PPM. Post-translational variable modifications consisted of methionine oxidation, asparagine and glutamine deamidation. Peaks studio built-in decoy sequencing and FDR determination was used to validate MS/MS based peptide and the parsimony rules for protein identifications. A threshold of the −10*logp (p-value) of 20 or greater was applied for the peptide identifications. The weighted sum of 9 parameters for peptide scoring are converted to a $p$-value which represent the probability of a false identification. Protein identifications were accepted if they could pass the −10logp of 20 and contained at least 1 identified unique peptide. Proteins that contained similar peptides and could not be differentiated based on MS/MS analysis alone were grouped to satisfy the principles of parsimony. Proteins sharing significant peptide evidence were grouped into protein groups. Final list generation was done by taking all proteins with a spectral count of five and above from each replicate reaction and comparing them in a venn-diagram against IgG control replicates. Proteins unique to both experimental replicates were then applied to the PANTHER database for protein ontology results.

## Co-immunoprecipitation and mass spectrometry

Fresh proteins were isolated from primary HESCs, PGR and TRIM28 overexpressed Hela cells and mouse uterus. The proteins were first incubated with 1 μg PR HA-tag TRIM28 or Rabbit IgG (supplementary data 6) at 4 °C overnight, then incubated with 40 μl Dynabeads™ Protein A at 4 °C for 3 h. After incubation, the beads were washed with lysis buffer for three times and boiled with 2× Laemmli Sample Buffer. The supernatant was collected for western blot. The immunoprecipitated proteins using TRIM28/rabbit IgG antibody in pre-decidual HESCs were run on polyacrylamide gel and submitted to the Mass Spectrometry Research and Support Group in NIEHS for mass spectrometry ($N = 1$ per antibody).

For mass spectrometry, each lane from the polyacrylamide gel was manually cut into 24 equal pieces. These pieces were then minced and placed into separate wells in a 96 well plate. Proteins in the gel pieces were digested with trypsin (Promega) for 8 hours using a Progest robotic digester (Genomic Solutions). Briefly, minced gel bands were incubated twice for 15 minutes in 100 μL of 25 mM ammonium bicarbonate, 50% (v/v) acetonitrile. The gel was then dehydrated by a 20-min incubation in 100 μL of acetonitrile followed by drying under a nitrogen stream. 250 nanograms of trypsin (Promega) were added, followed by an 8-h incubation at 37 °C. The resulting peptides were extracted by first collecting the digest supernatant, then incubating the gel with 50 μL of water for 20 minutes and collecting the resulting supernatant, and, finally, collecting the supernatant from 2 independent 20-min incubations of the gel in 50 μL of 5% (v/v) formic acid, 50% (v/v) acetonitrile. All supernatants for each well were pooled during the collection process. After the digestion process was complete, the 24 wells for each gel lane were batched and pooled by fours such each lane was condensed to 6 samples. These resulting samples were lyophilized to dryness. The lyophilized samples were resuspended in 25 μL of 0.1% formic acid prior to analysis.

Protein digests were analyzed by LC/MS on a Q Exactive Plus mass spectrometer (ThermoFisher Scientific) interfaced with a nanoAcquity UPLC system (Waters Corporation) equipped with a 75 μm × 200 mm HSS T3 C18 column (1.8 μm particle, Waters Corporation) and a Symmetry C18 trapping column (180 μm × 20 mm) with 5 μm particle size at a flow rate of 450 nL/min. The trapping column was positioned in-line of the analytical column and upstream of a micro-tee union which was used both as a vent for trapping and as a liquid junction. Trapping was performed using the initial solvent composition. A total of 5 μL of each peptide digest (6 samples per lane) were injected onto the column. Peptides were eluted by using a linear gradient from 99% solvent A (0.1% formic acid in water (v/v)) and 1% solvent B (0.1% formic acid in acetonitrile (v/v)) to 40% solvent B over 100 minutes. For the mass spectrometry a data dependent acquisition method was employed with a dynamic exclusion time of 15 s and also exclusion of singly charged ions. The mass spectrometer was equipped with a NanoFlex source and a stainless-steel needle and was used in the positive ion mode. Instrument parameters were as follows: sheath gas, 0; auxiliary gas, 0; sweep gas, 0; spray voltage, 2.7 kV; capillary temperature, 275 °C; S-lens, 60; scan range (m/z) of 375 to 1500; 1.6 m/z isolation window; resolution: 70,000; automated gain control (AGC), $3 \times 10e6$ ions; and a maximum IT of 100 ms. For the MS/MS scans: TopN: 10; resolution: 17500; AGC $5 \times 10e4$; maximum IT of 50 ms; and an (N) CE: 27. Mass calibration was performed before data acquisition using the Pierce LTQ Velos Positive Ion Calibration mixture (ThermoFisher Scientific). Peak lists were generated from the LC/MS data using Mascot Distiller (Matrix Science). The peak lists for the six independent injections that corresponded to a single lane were searched together using the Spectrum Mill software package (Agilent) against the Swissprot database (01_2014). Searches were performed using trypsin specificity and allowed for one missed cleavage and variable methionine oxidation. Mass tolerances were 20 ppm for MS scans and 50 ppm for MSMS scans. Proteins identified in TRIM28 immunoprecipitated samples were qualitatively compared to proteins found in IgG control samples to identify putative interactors of TRIM28.

## Proximal ligase assay of TRIM28 and PR in HESCs

After 48 h siNT or siTRIM28 knockdown followed by three-day EPC or vehicle treatment, the colocalization of PR and TRIM28 were detected by and TRIM28 antibody (Supplementary data 6) using Duolink® Proximity Ligation Assay, and visualized by the red fluorescence signaling under the confocal scope Zeiss-710. Primary HESCs from two different donors were checked in this assay.

## Luciferase report assay

HEC1A cells were cultured in McCoy's 5 A supplemented with 5% Fetal bovine serum (FBS) and 1× Penicillin Streptomycin (P/S). HEC1A cells were seeded at 24-well plate at 150,000 cell density and changed to McCoy's 5 A supplemented with 5% charcoal stripped Fetal bovine serum (FBS) for plasmid transfection. 100 ng TRIM28[76], PR, ERα and/or pcDNA3.1 vector plasmid were co-transfected with Ihh19 enhancer luciferase reporter plasmid[77] and pRL-TK plasmid by 1.2ul lipofectamine 2000 in 24 well plate. 24 h after transfection, the wells transfected with PR and three wells with only TRIM28 or only vector were treated with 10 nM R5020, while the wells transfected with ERα and three wells with only TRIM28 or only vector were treated with 10 nM 17β-estradiol for another 24 h. After 48 h of transfection, the cells were lysed and the luciferase were read following the manual of Dual-Luciferase® Reporter Assay System (Promega). $N = 3$ wells per group.

## 6-month breeding trial

8 weeks old virgin TRIM28[f/f], TRIM28[d/d], and eTRIM28[d/d] female mice were mated with stud C57BL/6 J males for 6 months. The date of delivery, the number of pups in each litter and the number of litters were recorded.

## Embryo transport, development and implantation during early pregnancy

A total of 8 weeks old virgin $Pgr^{cre/+}$, TRIM28$^{f/f}$, TRIM28$^{d/d}$, and eTRIM28$^{d/d}$ mice were mated with stud CD1 males. The morning that the mating plug observed was defined as D0.5. Embryo implantation was checked at D4.5 using retro-orbital 1% Evan's blue dye injection. The embryo development and transportation into the uterus were further analyzed by flushing the uterus and oviduct at D3.5.

Another set of $Pgr^{cre/+}$, TRIM28$^{f/f}$, TRIM28$^{d/d}$ and eTRIM28$^{d/d}$ female mice were mated and dissected at D3.5. The pregnancy status was determined by the presence of embryos in one side of the uterine horn and the oviduct through 1X PBS flushing. The flushed uterine horn was frozen on dry ice and stored at −80 °C for RNA isolation. The unflushed uterine horn and ovary was fixed in 4% PFA over-night at 4 °C.

## Serum hormone levels

The blood was collected from the retro-orbital veins of pregnant D3.5 mice. The pregnancy status was confirmed by the presence of embryos flushed from the uterus or oviduct. The blood was clotted at RT for 1 h, then centrifuged at 2000 g for 10 min. The clear supernatant was collected and sent to Virginia ligand core for mouse progesterone and 17β-estradiol ELISA tests. $N = 6$.

## Artificial decidualization and hormone treated mouse models

$Pgr^{cre/+}$, TRIM28$^{f/f}$, TRIM28$^{d/d}$ and eTRIM28$^{d/d}$ female mice were ovariectomized then rest for 2 weeks. The $Pgr^{cre/+}$, TRIM28$^{f/f}$, and TRIM28$^{d/d}$ mice were s.c. injected with 1 mg P4, 100 ng E2 or oil. The uterus was collected and frozen for RNA isolation after 6 h ($N = 4–6$). The TRIM28$^{f/f}$, TRIM28$^{d/d}$ and eTRIM28$^{d/d}$ female mice were treated with 100 ng E2 for three days, rest two days, 1 mg P4 + 6.7 ng E2 for three days. Oil was injected into one uterine horn to induce decidualization and further treated with P4 + E2 for 5 days, then weight each side of the uterus.

## Immunohistochemistry

A total of 5 μM thick paraffin embedded tissue sections were dewaxed and rehydrated for immunohistochemistry. Antigen retrieval were performed by boiling in the Antigen Unmasking Solution (H-3300, Vector laboratories) at 70% power for 3.5 min, and 20% power for 12 min using microwave. The slides were permeabilized by 0.2% Triton-100 for 10 min, and blocked by 3% hydrogen peroxide in methanol for 10 min and 5% normal donkey serum for 1 h at room temperature. Then they were incubated with specific primary antibody at 4 °C overnight (Supplementary data 6).

On the second day, the slides were incubated with biotinylated secondary antibody () for 1 h at room temperature, followed by the ABC reagent (PK-6100, Vector laboratories) for 1 h at room temperature. Signal was developed by DAB (SK-4105, Vector laboratories) for 30 s. The slides were counterstained with hematoxylin, dehydrated, cleared and mounted on Permount medium. The images were taken using Axiocam microscope camera (Zeiss).

## Immunofluorescence

Fixed four-chamber cell slides and dewaxed paraffin uterine section slides were boiled with Antigen Unmasking Solution as described above. After cooling at room temperature for 30 min, they were incubated with 0.2% Triton-100 for 10 min, 5% normal donkey serum for 1 h. Then slides were incubated with primary antibody at 4 °C overnight (Supplementary data 6).

On the second day, the slides were incubated with Alex Fluor secondary antibody (Supplementary data 6) for 1 h at room temperatures, then mounted with Vectashield Antifade Mounting Medium with DAPI (H1200, Vector laboratories). The images were taken under confocal microscope Zeiss-710.

## In situ hybridization

A total of 10 μm frozen sections of D3.5 TRIM28$^{f/f}$ and TRIM28$^{d/d}$ uterus were collected on slides, fixed with 4% PFA, dehydrated, incubated with RNAscope® Probe-Mm-Pgr, or Probe-Mm-Ihh for 3 h at 40 °C, amplified the signals using RNAscope® Fluorescent Multiplex Assay based on the manufacture's handbook, and the pictures were taken by confocal Zeiss 710 using 40× oil object within 24 h.

## Western blot

The proteins were loaded onto the 10% Mini-protein TGX stain free gels and separated at 150 volts. The protein was semi-dry transferred using the Trans-blot Turbo transfer system. The membrane was blocked with 5% non-fat dry milk for 1 h at room temperature and incubated overnight with primary antibody (Supplementary data 6). On the second day, the membrane was washed with 0.1% tween PBS 3 times for 5 min, and incubated with secondary antibody (Supplementary data 6) for 1 h at room temperature. The Amersham ECL Western blotting system was utilized to develop signal. The image was taken by Chemi-doc imager. The uncropped western images are at Supplementary data 7.

## RNA Isolation

The total RNA of the cells was isolated using Qiagen RNeasy RNA mini prep kit following the manufacturer's instructions.

The frozen uterus was homogenized in TRIzol reagent. The TRIzol solution was mixed with 1-Bromo-3-chloropropane. The aqueous phase was collected, mixed with chloroform and the aqueous phase was collected again. 75% ethanol was applied to the aqueous layer at 1:1 ratio, and the total solution was administered to the Qiagen RNEasy RNA mini prep kit column following the manufacturer's instructions.

## Real-time PCR

A total of 1 μg total RNA was reverse transcribed into cDNA using M-MLV reverse transcriptase according to manufacturer's instructions. Quantitative real time PCR was performed using SsoAdvanced Universal SYBR Green Supermix. SYBR green primers were designed using NIH Primer-blast or downloaded from the Primerbank[78] and synthesized by Sigma-Aldrich (Supplementary data 6). ΔΔCt values were calculated using 18 S control amplification results to acquire relative mRNA levels per sample.

## Assay for Transposase-Accessible Chromatin using sequencing (ATAC-Seq)

The primary HESCs treated with siNT or siTRIM28 for 48 h followed by EPC or vehicle treatment for 3 days. Cells from two different donors has been collected as biological replicates. In total, 8 cell samples were collected for ATAC-seq analysis according to the published protocol[79].

## Chromatin immunoprecipitation and sequencing (ChIP-Seq)

Pre-decidual and decidual HESCs were fixed with 1% PFA for 15 min, scraped from the dish and frozen on the dry ice. ChIP-seq has been performed on the chromatin collected from two different donors as biological replicates. Wildtype mouse uterus were collected at D3.5 and flash frozen on the dry ice. ChIP-seq has been performed on the chromatin collected from two batch of wildtype mice as biological replicates. Both the fixed HESCs and fresh frozen mouse uterus were sent to Active Motif for ChIP analysis using PR and TRIM28 antibody (Supplementary data 6).

## CUT&RUN

A total 0.5 M Decidual HESCs were incubated with PR or TRIM28 ((supplementary data 6) using the CUTANA™ ChIC/CUT&RUN Assays (Epicypher) following the manual. The library of captured DNA were amplified by Accel-NGS® 2 S DNA Library Kits and sequenced by Nextgen medium output.

## ChIP-qPCR and CUT&RUN qPCR

D3.5 TRIM28[f/f] and TRIM28[d/d] mouse uterus were fixed in 1% PFA for 10 min and quenched with 0.125 M glycine. The tissues were homogenized by Dounce homogenizer. The chromatin was isolated, sonicated into 200–600 bp, and immunoprecipitated using primary antibody. The chipped DNA as well as the input chromatin was heated for de-crosslinking, treated by RNase and Proteinase K and purified by QIAquick PCR Purification Kit for qPCR. The primers for the PR and ERα binding peaks were designed by NIH Primer-blast (Supplementary data 6). The Mouse Negative Control Primer Set 1 for ChIP-qPCR was purchased from Active motif. The Cq of PR binding peaks was first normalized by subtracting the Cq of the negative control primer. The enrichment fold was calculated by comparing the normalized Cq from the ERα ChIP to that of the input. $N = 6$ mice.

The DNA captured by PR CUT&RUN were diluted in the distilled water and used for qPCR. The sonicated cell chromatin was used for input. The primers were designed for PR binding peaks. The Human Negative Control Primer Set 1 was purchased from active motif. The enrichment fold was calculated as above. The primary HESCs from two donors were used in this assay.

## scRNA-Seq

Three control mice have been pooled as one TRIM28[f/f] sample, and four mutant mice have been pooled as one TRIM28[d/d] sample for scRNA-Seq. Single cell suspension were isolated using trypsin digestion. Briefly, the uteri were washed in ice cold PBS for three times, then incubated in 0.25% Trypsin -EDTA on ice for 1 h, then at RT for 10 min, and 37 °C for 50 min. During the 37 °C incubation, the uteri were hand-shook gently every 5 min. The cell pellets were obtained by centrifuge at 450 g for 10 min at 4 °C. Then the cell suspension was filtered through 100 μm and 70 μm mesh sequentially. The blood cells and debris were removed by red blood cell lysis kit (130-094-183, Miltenyi Biotec).

## ATAC-seq, RNA-seq, ChIP-seq and scRNA-seq analysis

The fastq files were trimmed, mapped and deduplicated as we showed in our published code. The differentiated expressions of genes (DEGs) between siNT and siTRIM28 groups in the decidual or pre-decidual primary HESCs from three donors were determined using the paired wise analysis of EdgeR[80]. The DEGs between D3.5 TRIM28[f/f] and TRIM28[d/d] mouse uterus were calculated using Cufflinks[81]. The DEGs were set as mean FPKM ≥1 in at least one of the conditions; fold change of >1.5 (up-regulated) or <−1.5 (down-regulated) for primary HESCs and fold change of >2 (up-regulated) or <−2 (down-regulated) for mouse uterus; and p-value < 0.05.

The signatures of siTRIM28 in decidual HESCs and TRIM28 deletion in mouse uterus were determined as the DEGs with fold change of >2 (High) or <−2 (Low). The correlations of the gene signatures with the published human endometriosis dataset GSE141549[55] were predicted by T score using the SEMIP R package[82].

The expressions of DEGs were normalized and scaled, then a heatmap was generated using the intensity plot tool of Partek Genomics Suite 6.6. The significantly enriched upstream regulators of the DEGs were identified using Ingenuity pathway analysis. The DEGs in this study were correlated with other published studies using Illumina Nextbio. The Pearson correlation coefficient of the DEGS in the pre-decidual and decidual HESCs were calculated by Graphpad Prism 9.

For ChIP-Seq, the retained read alignments were sorted by coordinates, extended to 300 bases, and peaks were called using MACS2[79] with FDR cutoff at 0.0001. For cut and run, the uniquely mapped reads in each sample were down to 10 million. The peaks were identified using MACS2 with a cutoff of adjusted p-value less than 0.01 for PR and 0.1 for TRIM28. The peak overlap was defined as 1 bp overlap. The peak associated genes were annotated based on its direct overlap with the gene regulatory domain by GREAT using the basal plus extension setting[83]. For the peaks in mouse uterus, the peak overlapped HiC loops have also been used to predict the related gene using GREAT. Motif analysis used Homer[84]. The heatmap of ChIP-Seq and ATAC-seq peaks were generated using the webtool EaSeq (http://easeq.net)[85].

scRNA-seq analysis followed the 10× genomic work flow and the Seurat[86] package as showed in the published code. After filtering by mitochondria genes <25% and number of gene features 200–7500, 5729 cells from TRIM28[f/f] and 4584 cells from TRIM28[d/d] were used for the subsequent analysis. Monocle[44] and RNA velocity[45] has been used for cell trajectory inference.

## Statistical analysis

The Realtime PCR results, the number of proliferative and migrated cells, delivered pups, embryo implantation sites, the ratio of injected over un-injected uterine weight and serum estradiol and progesterone levels were analyzed by Shapiro-wilk test to validate its normality then compared by student's test, two sided. For multiple group analysis, such as luciferase results, T score and qPCR of siNT or SiTRIM28 treated decidual and pre-decidual HESCs, one-way ANOVA with post-hoc Turkey were used for comparison. The percentage of abnormal embryo and embryo in the oviduct were compared by Fisher's exact test. All data graph with statistical analysis presented as dot plot with median labeled as line. *$p < 0.05$. The individual data for all the dotplot are listed at Supplementary data 8.

## Reporting summary

Further information on research design is available in the Nature Portfolio Reporting Summary linked to this article.

## Data availability

All the raw and processed data of RNA-Seq, ATAC-Seq, ChIP-Seq and scRNA-Seq were deposited to GSE205481. Published mouse uterine H3K27AC, PR ChIP-seq (GSE178542, https://www.ncbi.nlm.nih.gov/geo/query/acc.cgi), mouse ERα ChIP-Seq (GSE36455, https://www.ncbi.nlm.nih.gov/geo/query/acc.cgi), mouse HiC (GSE147843, https://www.ncbi.nlm.nih.gov/geo/query/acc.cgi), human endometrial ERα (GSE200807, https://www.ncbi.nlm.nih.gov/geo/query/acc.cgi) and PR (GSE132713, https://www.ncbi.nlm.nih.gov/geo/query/acc.cgi) ChIP-Seq were downloaded from GEO. The mass spec data of pre-decidual HESCs were deposited to MassIVE (https://massive.ucsd.edu/ProteoSAFe/dataset.jsp?accession=MSV000092346). All the analysis files for RIME in decidual HESCs were deposited to figshare as https://doi.org/10.6084/m9.figshare.23687868. All the raw pictures for this paper were deposited to figshare as https://doi.org/10.6084/m9.figshare.23688651.

## Code availability

All the code were deposited to https://github.com/lirongdme/TRIM28_paper.git.

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

## Acknowledgements
This work was supported by an Intramural Research Program of the NIEHS, NIH project nos. Z1AES103311 (F.J.D.) and NIH/NICHD R01 HD-042311 (J.P.L.). We acknowledge the Epigenomic and DNA Sequencing Core, The DNTP Clinical Pathology Core, the Integrative Bioinformatics Supportive Group, the Fluorescence Microscopy and Imaging Center, the Mass Spectrometry Research and Support Group and the Comparative Medicine Branch at NIEHS for their research support. The authors acknowledge Research in Reproduction Ligand Assay and Analysis Core, University of Virginia for serum hormone analysis. We appreciated Mr. Linwood Koonce for mouse colony management, Ms. Mita Ray for preparing the reagent.

## Author contributions

R.L., S.W., and F.J.D. designed the experiments. R.L., J.P.L., S.W., and F.J.D. wrote the paper. R.L. and R.M.M. performed the experiments. R.L. and T.W. conducted the bioinformatics analysis.

## Funding

## Competing interests

The authors declare no competing interests.
