## [Peer Review File · Nature Communications]

TRIM28 modulates nuclear receptor signaling to regulate uterine functionReviewer #1 (Remarks to the Author):

The authors in the current study showed that Trim28 is expressed in different uterine cells. The ablation of Trim28 causes compromised decidualization in human uterine fibroblasts. The conditional deletion of Trim28 using either Pgr-cre and Ltf-cre results in pregnancy failure. In Trim28 deficient uteri, the authors identified a group TRIM28 positive and PGR negative pericytes and epithelial cells with progenitor potential. The study demonstrated Trim28 plays a key role in progesterone and estrogen signaling.

This study involves multiple mouse lines and cell models, as well as tremendous high throughput data. I am really impressed by the amount of work the authors have completed. They tried to tackle the question from different angles including hormonal signaling pathways, uterine cell composition, and potential uterine progenitor cells. In the meanwhile, this raises a major concern that many of the conclusions were drawn by high-throughput data analysis. One of the major conclusions is that Trim28 forms a complex with Pgr and Esr to influence Pgr/Esr function. However, the paper showed physical association of the components by protein IP, and potential association by ChIP binding peaks. The functional coupling of Trim28 and Pgr/Esr has not been tested.

Figure 1B, nuclear localized PGR is very low in decidualized cells. Could the authors explain the potential reasons? It's not surprising that two nuclear proteins (Trim28 and PGR) are all localized in nuclei, although the author would like to show the co-localization of the two proteins.

The quality of figure 1C is not high. Although the yellow frame highlighted some bright red dots, many other similar bright red dots are observed outside of cell nuclei, suggesting a high background. It greatly weakens the credibility of figure 1C.

The broad expression of Trim28 makes it difficult to explain the data. Although the authors claimed that murine Trim28 is required for endometrial decidualization, the evidence to support Trim28's role in decidualization in vivo is not enough. Given the implantation failure in Trim28f/fPgrCre/+ females, it is possible the luminal epithelial deletion of Trim28 causes decidual defects. The artificial decidualization also requires functional luminal epithelia to transmit the stimuli to stromal cells.

In Figure 4, the authors hypothesized that Trim28 ablation may alter PGR chromatin binding activity. However, the authors just give an example of binding peaks around Ihh. It could barely provide an overall view of PGR chromatin binding. In addition, the data provided association evidence but not direct and functional coupling between the peaks and Ihh expression.

The pseudo-time analysis is not enough to support the conclusion that Mustn1 pericyte cluster had progenitor characteristics.

It is not appropriate to use the subtitle: TRIM28 positive PGR negative progenitor cells in the uterine epithelial compartment. Single cell analysis in Trim28f/f and d/d uterine cells showed different cell clusters. The enrichment of progenitor genes is not enough to claim progenitor cells.

Some minor points:

5c, no peak intensity.

Line 95, the labeling "(Fig. 1B)" is wrong.

Line 98, the labeling "(Fig. 1C)" is wrong.

Line 166, the labeling "(Fig. 2A, S3B)" is wrong.

Line 175, the state "TRIM28 d/d uterus ...exhibited limited ...decidual marker HAND2 expression" is not supported by figure 3A.

In the legend of figure 3, the last "D." should be "E."

Line 316. This sentence is not clear: "At 8-week-old TRIM28 was undetectable in the TRIM28d/d uterine stromal cells in which PGR levels were low (Fig. 7A and S6A). Surprisingly, TRIM28 immunopositivity was detected in a subset of stromal cells"

Reviewer #2 (Remarks to the Author):

The authors perform RIME experiments to identify PGR interactors in decidualized uterine HESCs. They identify TRIM28 to be an important chromatin binder that interacts with both PR and ER and affects hormone signaling pathways in the uterus. They show that TRIM28 knockdown through siRNA technology or the knock-out in genetic mouse models results in implantation defects due to its deleterious effects on endometrium regulation. They reveal TRIM28 to be a possible non hormonal target for treatment of endometriosis and other systemic diseases.

Overall, the scientific findings are mostly sound. However, the manuscript lacks details on a experiments, replicates, analyses used.

Major points

1. For ChIP-seq, ATAC-seq and scRNAseq, number of replicates is usually not stated. Unless there is strong orthogonal data confirming the 'seq' finding, a minimum of two replicates are required. As much as I think the data looks really interesting, a single replicate is not sufficient.
2. The manuscript is lengthy and difficult to follow. I would strongly suggest reducing main figures to focus only on key findings, moving anecdotal findings to supplementary. For example, a quarter of figure 1 is dedicated to showing co-ip results, but then Fig6A shows similar results again, with an extra target. These can be summarized in one figure.

Minor points

- Most figure legends could use more explanation and clarity. When showing results from multiple experiment types, each experiment/results needs to be a sub panel. For example, Fig4 only has 4a-d but has over 10 data panels. This makes it difficult for a reader to follow the text and understand.
- Pay attention to using ESR1/PGR to refer to the genes and ER α /PR to refer to the proteins/receptors
- use Trim28 when talking about the mouse gene and TRIM28 for human gene.
- line 166: Figure 3A is referred to as Figure 2A.
- Figure 3E is referred to as D in the text. Also, the scale bars have shifted on the figure.
- Figure 2B could be better labelled. Which direction is the KD?
- Figure 3D, numbers in venn diagram need aligning.
- Overall, the paper does not describe the number of DEG or chip-peaks identified for a given experiment and instead show percentages. I think its important you highlight the numbers.
- Figure 7C perceived direction of pathway change and text don't match. Perhaps the figure could be labelled to indicate direction change.
- Figure 7C. What specific analysis was done here? What do circle size and color denote?
- What pseudo time analysis was performed? There results don't look particularly convincing. If you haven't already tried this, i would suggest analysis that compare spliced/unspliced ratios, such as Velocity.
- Figure 5C legend: again, what's the basis for the blue lines? How are these contacts calculated?
- Figure 6A, what is IB? clarify in legend
- Authors refer to other datasets being used without referencing them in lines 309 and 413.
- Correct l. 448 to "condensate" instead of condense.
- On l. 326 what do you mean by "pooled"? If they are run on the same lane, how did you differentiate between TRIM28 fl/fl and d/d cells.
- RIME results need to be explained. How many replicates, what was the control. Also, full list of proteins identified was not shown (with peptide counts etc) as far as I could see.
- Corrections needed on l. 194, l. 235-238, l. 351

The authors are very grateful for the reviewer's careful review and insightful suggestions. The authors have taken the reviewers comments and significantly revised the manuscript and the incorporation of these comments have improved the paper quality. We addressed all the questions in the re-submitted draft and listed the responses to each reviewer's question as below.

Reviewer #1 (Remarks to the Author):

The authors in the current study showed that Trim28 is expressed in different uterine cells. The ablation of Trim28 causes compromised decidualization in human uterine fibroblasts. The conditional deletion of Trim28 using either Pgr-cre and Ltf-cre results in pregnancy failure. In Trim28 deficient uteri, the authors identified a group TRIM28 positive and PGR negative pericytes and epithelial cells with progenitor potential. The study demonstrated Trim28 plays a key role in progesterone and estrogen signaling.

This study involves multiple mouse lines and cell models, as well as tremendous high throughput data. I am really impressed by the amount of work the authors have completed. They tried to tackle the question from different angles including hormonal signaling pathways, uterine cell composition, and potential uterine progenitor cells. In the meanwhile, this raises a major concern that many of the conclusions were drawn by high-throughput data analysis.

Comment 1: One of the major conclusions is that Trim28 forms a complex with Pgr and Esr to influence Pgr/Esr function. However, the paper showed physical association of the components by protein IP, and potential association by ChIP binding peaks. The functional coupling of Trim28 and Pgr/Esr has not been tested.

Response 1: In order to test the functional coupling between TRIM28 and PGR/ESR1 we determined the direct regulatory effects of TRIM28 on PGR/ESR1 transcription activity using luciferase reporter assay.

We found PGR, ESR1 and TRIM28 had overlapped peaks at *Ihh* 19kb enhancer of mouse uterus which has been showed crucial for *Ihh* regulation during pregnancy¹. These regulatory regions were also found at the upstream of *IHH* in human endometrium. Thus, we take advantage of the luciferase reporter plasmid that we have designed before² to determine the direct regulatory effects of TRIM28 on mouse *Ihh* 19kb enhancer.

We transfected the Human endometrial cancer cell (HEC1A) cells with the *Ihh* 19kb enhancer reporter plasmid, pRL Renilla control plasmid, human PGR/ESR1 and/or TRIM28 plasmid to check the relative luciferase intensity. In mouse uterus, we already showed that TRIM28 deletion reduced PGR binding activity at *Ihh* 19kb enhancer and decreased *Ihh* expression. From the luciferase assay, we found overexpression of human PGR isoforms A or B, or human ESR1 can induce luciferase intensity under progesterone or estrogen treatment, and the co-transfection of TRIM28 further enhanced the induction. These results indicate that TRIM28 can directly activate the PGR and ESR1 transcription activity.

The results were shown at Fig. 6H, supplementary Fig. 7E-F and line 321-326:

"To directly unveil the functional coupling between TRIM28 and PR/ER α , we performed luciferase reporter assay in Human endometrial cancer cell line (HEC1A). We found TRIM28

can further enhance two PR isoforms, PRA/PRB and ER α transcription activity at the 19kb upstream enhancer of *Ihh* (Fig. 6E, Supplementary Fig. 7E-F), indicating that TRIM28 directly facilitate the transcriptional regulatory functions of PR/ER α ”

Supplementary Fig. 7E-F

Fig. 6E

Comment 2: Figure 1B, nuclear localized PGR is very low in decidualized cells. Could the authors explain the potential reasons? It's not surprising that two nuclear proteins (Trim28 and PGR) are all localized in nuclei, although the author would like to show the co-localization of the two proteins.

Response 2: The reason that only a small subset of cells showed high levels of nuclear PGR expression is because of the dynamic expression patterns of PGR in the hormone treated human cell lines³⁻⁵.

Specifically, PGR is diffused at the cytoplasm and nucleus with a half-life time of 21h without the stimulation of hormone. Once the hormone added, PGR accumulates at the nucleus within 1h to promote the transcription regulation followed by rapid down regulation with a half-life time around 6h. The hormone-dependent PGR downregulation is caused by both increased PGR protein degradation and decreased *Pgr* transcription.

Therefore, it is expected that after three-day hormone treatment, PGR will not be detected in most of the decidualized human endometrial stroma cells. On the contrary, PGR is most likely to be transiently expressed by a subset of decidual stromal cells to mediate their

decidualization process at one time. In this case, as we snapshot the cells using IF of Fig. 1B, we only detected a small subset of cells showed nuclear expression of PGR.

In the paper, we describe the figure at line 93-95:

“In a subset of decidualized HESCs, TRIM28 and PR were co-expressed in the nucleus in close proximity (Fig. 1B, C).”

Comment 3: The quality of figure 1C is not high. Although the yellow frame highlighted some bright red dots, many other similar bright red dots are observed outside of cell nuclei, suggesting a high background. It greatly weakens the credibility of figure 1C.

Response 3: We agreed with the reviewer the Duolink Proximity Ligation Assay has some non-specific background which is most likely caused by non-specific binding of the primary or secondary antibody. Therefore, to differentiate the signal and background, we included the siTRIM28 treated HESCs as the negative control to validate the results. As expected, the TRIM28 and PGR proximity labeling were dramatically reduced in the siTRIM28 treated HESCs, which indicated that the bright staining we observed in the siNT group was most likely the real signals.

It is also intriguing to us that there are some strong staining seems located out of the nucleus. One possibility is TRIM28 and PGR is also proximally located in the cytoplasm. Although the levels are much lower, TRIM28 and PGR can be found at the cytoplasm to mediate cell signals^{6,7}. We are hoping that techniques with better sensitivity and specificity can be developed in the future to explore the cytoplasmic TRIM28-PGR interactions and our current study can provide some insight.

Fig. 1C

Comment 4: The broad expression of Trim28 makes it difficult to explain the data. Although the authors claimed that murine Trim28 is required for endometrial decidualization, the evidence to

support Trim28's role in decidualization in vivo is not enough. Given the implantation failure in Trim28^{f/f}/PgrCre/+ females, it is possible the luminal epithelial deletion of Trim28 causes decidual defects. The artificial decidualization also requires functional luminal epithelia to transmit the stimuli to stromal cells.

Response 4:

We agreed with the reviewer that epithelial TRIM28 is critical for the decidualization in mouse uterus. Through study about primary human endometrial stromal cells, we found the crucial roles of TRIM28 in stromal cells. But based on two uterine specific TRIM28 knockout mouse models, we further identified indispensable roles of TRIM28 in the uterus including stroma and epithelium.

In the uterine specific *Pgr*^{Cre/+}*Trim28*^{f/f} mice, TRIM28 has been deleted from both epithelium and stroma. We already found TRIM28 deletion impaired the uterine epithelium functions through the dysregulation of multiple epithelial specific genes such as *Ihh*, *Lif*, and *Ltf*. Actually, *Lif* and *Ihh* has already been reported as the critical epithelial regulator of uterine decidualization^{8,9}. Furthermore, we found the TRIM28 deletion suppressed PGR occupancy at the *Ihh* enhancers to reduce *Ihh* expression. Therefore, the uterine specific TRIM28 knockout mouse, *Pgr*^{Cre/+}*Trim28*^{f/f}, already implied the epithelial contribution to the decidualization defects.

In order to get a clear conclusion of the reviewer's comments, we performed artificial decidualization in the epithelial specific *Ltf*^{Cre/+}TRIM28^{f/f} mice. We found epithelial deletion of TRIM28 caused decidualization failure (Supplementary Fig. 6D) and disrupted the expressions of multiple epithelial specific genes including decreased *Lif* and *Ihh* (Supplementary Fig. 6E). All these results indicate epithelial TRIM28 is critical for decidualization.

Meanwhile, by comparing uterine specific *Pgr*^{Cre/+}*Trim28*^{f/f} females with epithelial specific *Ltf*^{Cre/+}TRIM28^{f/f} TRIM28 knockout mice, we observed several epithelial independent phenotypes, such as the reduced stromal PGR, stroma cell proliferation and vascular density, as well as the abnormally accumulation of PGR-TRIM28+ stromal cells in the *Pgr*^{Cre/+}*Trim28*^{f/f} uterus (Fig. 3B, C, F, 7A). Therefore, these stromal defects were most likely resulted from the stromal specific TRIM28 deletion. More importantly, these stromal defects including stromal PGR expression¹⁰, stromal cell proliferation¹¹ and angiogenesis¹² have been regarded as the crucial regulator or biological process for decidualization.

Among them, the reduced proliferation and suppressed PGR signaling in the mouse uterus is consistent with what we observed in the human endometrial stromal cells (Fig. 2B, C, D, Supplementary Fig. 1E). Besides, the scRNA-seq we performed in the *Pgr*^{Cre/+}*Trim28*^{f/f} uterus further verified several conserved and distinct pathways altered between epithelium and fibroblasts that may play important roles in decidualization. Therefore, the stromal TRIM28 also plays important roles in the decidualization of mouse uterus.

In summary, our *in vivo* mouse models not only revealed the important roles of stromal TRIM28 in decidualization which is consistent to human endometrial stromal cells, but also indicated the indispensable roles of epithelial TRIM28 in stromal decidualization.

The artificial decidualization results of *Ltf*^{Cre/+}TRIM28^{f/f} mice were added at Supplement Fig. 6D and the dysregulated epithelial genes and unaltered stromal genes were at Supplement Fig. 6E.

We changed the sentence at line 260-263:

“No detected embryo implantation at D4.5 uterus with the blastocysts in the uterus and failed uterine responses during artificial decidualization suggested epithelial TRIM28 is also critical for decidualization (Supplementary Fig. 6B-D).”

We added the sentence at line 276-280:

“Interestingly, eTRIM28d/d mice did not show reduced stromal cell proliferation or decreased PR and PR/ERα target gene expression at the stroma which have been observed in the TRIM28d/d mice and are essential for decidualization^{16,39} (Supplementary Fig. 6F, G) suggesting that stromal and epithelial TRIM28 play critical and distinct roles in uterine biology.”

Supplementary Fig 6D

Supplementary Fig 6E

Comment 5: In Figure 4, the authors hypothesized that Trim28 ablation may alter PGR chromatin binding activity. However, the authors just give an example of binding peaks around *Ihh*. It could barely provide an overall view of PGR chromatin binding. In addition, the data provided association evidence but not direct and functional coupling between the peaks and *Ihh* expression.

Response 5: In addition of *Ihh*, we further validated the PGR chromatin binding activity at other 14 PGR-TRIM28 coregulated genes in control and *Pgr^{Cre/+}Trim28^{fl/fl}* uterus using ChIP-qPCR. These 14 genes were differentiated expression between control and *Pgr^{Cre/+}Trim28^{fl/fl}* uterus. And surrounding these genes there are at least one PGR and TRIM28 overlapped peaks that are located either at the promoter or within one chromatin loop.

Eventually, we found PGR binding activity was significantly reduced at the upstream of *C2cd4a*, *Muc20*, *Igf1*, promoter of *Rims1*, *Tac2*, *Nr4a3*, *Npl*, *Abca4*, *Ttr*, *Wfdc*, the downstream of *Msx2*, *Ttr*. However, PGR binding activity at *Calca* promoter and *Meig1* was not significantly altered. In this way, we think that the deletion of TRIM28 inhibited the PGR binding activity at multiple chromatin loci but not all of them. These results also suggested that the TRIM28 regulation on PGR may involve other players that is worth further study.

The functional coupling between TRIM28 and PGR/ESR1 were replied in our response to **comment 1**.

The results of added ChIP qPCR were mentioned at supplementary Fig. 5, line 247-250:

“Additionally, we found the PR binding activity has also been reduced in the TRIM28d/d uteri at multiple but not all the chromatin loci that are close to other PR-TRIM28 co-regulated genes (Supplementary Fig. 5) suggesting a gene specific regulatory role of TRIM28 on PR transcription activity.”

Supplementary Fig. 5

Comment 6: The pseudo-time analysis is not enough to support the conclusion that *Mustn1* pericyte cluster had progenitor characteristics. It is not appropriate to use the subtitle: TRIM28 positive PGR negative progenitor cells in the uterine epithelial compartment. Single cell analysis in *Trim28^{f/f}* and *d/d* uterine cells showed different cell clusters. The enrichment of progenitor genes is not enough to claim progenitor cells.

Response 6: We agree the pseudotime analysis is not enough evidence. The title is changed at line 378 to "*The characteristics of TRIM28 positive PR negative cells in the uterine stroma*".

Some minor points:

Comment 7: 5c, no peak intensity.

Response 7: The peak intensity has added at 5C and other genome browser screen shot.

Comment 8: Line 95, the labeling "(Fig. 1B)" is wrong.

Response 8: "Fig. 1B" was corrected as "Fig. 1B, C".

Comment 9: Line 98, the labeling "(Fig. 1C)" is wrong.

Response 9: "Fig. 1C" was corrected as "Fig. 1D".

Comment 10: Line 166, the labeling "(Fig. 2A, S3B)" is wrong.

Response 10: "Fig 2A, S3B" was corrected as "Fig. 3A, Supplementary Fig. 3C" at line 172.

Comment 11: Line 175, the state "TRIM28 *d/d* uterus ...exhibited limited ...decidual marker HAND2 expression" is not supported by figure 3A.

Response 11: HAND2 was ubiquitously expressed in almost all the stromal cells of *TRIM28^{f/f}* uterus indicating successful decidualization. But HAND2 was only expressed in a subset of sub-epithelial stromal cells indicating impaired decidualization.

Therefore, the sentence was changed at line as 176-178:

"The stimulated uterine horn in the TRIM28^{d/d} uterus failed to increase in the size and weight and exhibited fewer decidual marker HAND2 positive cells"

Comment 12: In the legend of figure 3, the last "D." should be "E."

Response 12: Due to the re-organization of the main and supplementary figure, the last in the legend of figure 3 was changed to F.

Comment 13: Line 316. This sentence is not clear: "At 8-week-old *TRIM28* was undetectable in the *TRIM28^{d/d}* uterine stromal cells in which PGR levels were low (Fig. 7A and S6A). Surprisingly, *TRIM28* immunopositivity was detected in a subset of stromal cells"

Response 13: Because the majority of uterine cells expressed PR, so we used *Pgr^{cre/+}* to delete TRIM28. IHC showed TRIM28 expression was lost in the PR expressing cells, but a large number of PR non-expressing cells abnormally accumulated in the uterus and expressed TRIM28.

The sentence now at line 332-336 was changed to:

“At 8-week-old TRIM28 was deleted in the TRIM28d/d uterine stromal cells which had low but detectable PR expressions (Fig. 7A and Supplementary Fig. 8A). Surprisingly, TRIM28 immunopositivity was still preserved in another subset of stromal cells in which PR protein was not detected (Fig. 7A and Supplementary Fig 8A).”

Reviewer #2 (Remarks to the Author):

The authors perform RIME experiments to identify PGR interactors in decidualized uterine HESCs. They identify TRIM28 to be an important chromatin binder that interacts with both PR and ER and affects hormone signaling pathways in the uterus. They show that TRIM28 knockdown through siRNA technology or the knock-out in genetic mouse models results in implantation defects due to its deleterious effects on endometrium regulation. They reveal TRIM28 to be a possible non hormonal target for treatment of endometriosis and other systemic diseases.

Overall, the scientific findings are mostly sound. However, the manuscript lacks details on a experiments, replicates, analyses used.

Major points

Comment 1: For ChIP-seq, ATAC-seq and scRNAseq, number of replicates is usually not stated. Unless there is strong orthogonal data confirming the 'seq' finding, a minimum of two replicates are required. As much as I think the data looks really interesting, a single replicate is not sufficient.

Response 1:

We agreed with the reviewer. For primary human endometrial stromal cells, the two biological replicates from two different donors have been used for H3K27AC, H3K27me3, TRIM28 ChIP-seq, PR, TRIM28, IgG CUT&RUN, and siNT or siTRIM28 treated ATAC-Seq. Conserved peaks from both patients are used for further analysis. Except CUT&RUN analysis has much lower peak number, so combined peaks from two biological replicates have been used for further analysis.

For mouse uterus, pooled wildtype mouse uteri chromatin has been used for ChIP-Seq analysis. Two biological replicates of TRIM28 and PR ChIP-seq have been performed separately. Conserved peaks were used for following analysis.

For scRNA-seq, three control mice have been pooled as one control sample, and four mutant mice have been pooled as one mutant sample to avoid the bias from individual mouse. After filtering by mitochondria genes <25% and number of gene features 200-7500, 5729 cells from control and 4584 cells from mutant were used for the subsequent analysis.

And the raw and processed files for all the biological replicates have been deposited to GSE205481.

Additionally, the H3K27AC ChIP-seq in D3.5 mouse uterus, PR ChIP-Seq and HiC in P4 treated mouse uterus, ER α ChIP-Seq and HiC in E2 treated mouse uterus, PR and ER α ChIP-seq in human endometrium are collected from published datasets and the original paper is cited in the correspondent text.

The description of biological replicates was added at method:

Line 692-693: "*ChIP-seq has been performed on the chromatin collected from two different donors as biological replicates.*"

Line 694-695: "*ChIP-seq has been performed on the chromatin collected from two batch of wildtype mice as biological replicates.*"

Line 701-702: "*CUT&RUN has been performed on the cells collected from two different donors as biological replicates.*"

Line 720-721: "*Three control mice have been pooled as one TRIM28f/f sample, and four mutant mice have been pooled as one TRIM28d/d sample for scRNA-Seq.*"

Supplementary method, Line 20: "*Cells from two different donors has been collected as biological replicates.*"

Comment 2: The manuscript is lengthy and difficult to follow. I would strongly suggest reducing main figures to focus only on key findings, moving anecdotal findings to supplementary. For example, a quarter of figure 1 is dedicated to showing co-ip results, but then Fig6A shows similar results again, with an extra target. These can be summarized in one figure.

Response 2:

We moved several subpanels from Fig 2-8 to supplement figures to include the major points in the paper.

In this way, **Fig.1** has four major points: RIME results to imply the potential interactions between PR and TRIM28, validation of PR and TRIM28 expression by IF, validation of PR and TRIM28 co-localization by PLA, validation of PR and TRIM28 interactions by co-IP.

Fig.2 has three major points: siTRIM28 treatment impaired HESC cell proliferation, migration and decidualization; TRIM28 and PGR shared conserved transcriptome and cistrome; siTRIM28 knockdown diminished PR binding activity at PR-TRIM28 co-regulated genes.

Fig. 3 has four major points: TRIM28 deficient mice had failed embryo implantation and artificial decidualization in which disrupted uterine biology is already obvious at pregnancy day 3.5; Top altered decidualization related pathways in TRIM28 deficient uterus; Suppressed PR signaling in TRIM28 deficient uterus; PR mRNA and Protein expression unaltered at uterine epithelium but decreased at stroma.

Fig. 4 has three major points: Suppressed Epithelial PR signaling; decreased PR binding activity at the enhancers of epithelial specific regulator *Ihh*; Epithelial specific TRIM28 knockout mouse can also reduce *Ihh* expression.

Fig. 5 has three major points: ER α protein levels unaltered; ER α and PR co-regulated genes and pathways; decreased ER α binding activity at the promoter of *Pgr*.

Fig. 6 has five major points: TRIM28 can co-IP with PR and ER α in mouse uterus; TRIM28, PR and ER α shared conserved chromatin binding patterns in mouse uterus; after ovariectomy, both PR and ER α signaling were inhibited in TRIM28 deficient mouse uterus upon progesterone or estrogen treatment; TRIM28, PR and ER α shared conserved chromatin binding patterns in human endometrium; TRIM28 co-transfection with PR and ER α can enhance the PR and ER α induced luciferase intensity.

Fig. 7 has three major points: TRIM28+/PR- cells accumulated in mouse stroma; scRNA-seq identified multiple cell clusters including the clusters that are TRIM28+/PR-; The characters of TRIM28-/PR+ and TRIM28+/PR- in the mouse stroma.

Fig. 8 has three major points: TRIM28+/PR- cells accumulated in mouse epithelium; TRIM28+/PR- epithelium has higher levels of Lgr5; LGR5 expression increased in the TRIM28 mutant epithelium.

Fig. 9 has one major point: TRIM28 deficient gene signatures in mouse uterus and human endometrial stromal cells are positive correlated with several human endometrial pathology especially endometriosis.

For Fig.1D and Fig.6A, we think current sequence is easier for readers to understand. Because false positive is commonly observed in RIME experiment, so we performed co-IP in Figure 1D to validate human PR and TRIM28 binding activity.

From Fig. 3-5, we noticed in the mouse uterus, TRIM28 may affect the transcription activity of both ER α and PR. So in Fig. 6A we further validated in pregnant mouse uterus, TRIM28 can also co-IP with ER α and PR, suggesting the *in vivo* interactions among mouse TRIM28, ER α and PR.

Therefore, the current separate presentation may be better than show them together.

To make our points clearer, at line we added:

“Moreover, the immunoprecipitation (IP) assay using the PGR antibody successfully pulled down TRIM28 in decidual HESCs further validated the RIME results”

At line 304-306 we added:

“Similar to the decidual HESCs, we found TRIM28, ER α and PR can form a complex in the D3.5 mouse uterus (Fig. 6A) indicating the conserved co-regulation of these proteins in both human and mouse.”

Minor points

Comment 3: Most figure legends could use more explanation and clarity. When showing results from multiple experiment types, each experiment/results needs to be a sub panel. For example, Fig4 only has 4a-d but has over 10 data panels. This makes it difficult for a reader to follow the text and understand.

Response 3: The figures were divided into more sub-panel to make sure each panel has one theme.

Comment 4: Pay attention to using ESR1/PGR to refer to the genes and ER α /PR to refer to the proteins/receptors

Response 4: Corrected. *ESR1/PGR* is used for the genes and ER α /PR is used for the protein.

Comment 5: use Trim28 when talking about the mouse gene and TRIM28 for human gene.

Response 5: *Trim28* is used for mouse gene, *TRIM28* is used for human gene, and TRIM28 is used for mouse and human protein.

Comment 6: line 166: Figure 3A is referred to as Figure 2A.

Response 6: Now Line 168 "Fig. 2A" Changed to "Fig. 3A".

Comment 7: Figure 3E is referred to as D in the text. Also, the scale bars have shifted on the figure.

Response 7: "Fig. 3D" Changed to "Fig. 3G" as we increased the sub panel in the figure. The scale was re-located.

Comment 8: Figure 2B could be better labelled. Which direction is the KD?

Response 8: A title "Top altered pathways by TRIM28 Knockdown" was added in Fig. 2B; In the figure legend, added "Green means inhibition. Red means activation".

Comment 9: Figure 3D, numbers in venn diagram need aligning.

Response 9: Number in Fig. 3D is aligned.

Comment 10: Overall, the paper does not describe the number of DEG or chip-peaks identified for a given experiment and instead show percentages. I think its important you highlight the numbers.

Response 10: The actual number of DEG and chip-peaks were presented in the text and figure.

Comment 11: Figure 7C perceived direction of pathway change and text don't match. Perhaps the figure could be labelled to indicate direction change.

Response 11: Now Fig. 7D. "*Top altered pathways in the Trim28 deleted fibroblast clusters*" was added in the figure. "*Green means inhibition. Red means activation*" was added at the figure legend.

Comment 12: Figure 7C. What specific analysis was done here? What do circle size and color denote?

Response 12: Now Fig. 7E. "*Enriched pathway in Mustn1 Peri annotated by IPA based on the cluster specific marker genes. The size of the red dot is proportional to the -logP value. The red*

color is proportional to the z score" was added to the figure legend.

Comment 13: What pseudo time analysis was performed? There results don't look particularly convincing. If you haven't already tried this, i would suggest analysis that compare spliced/unspliced ratios, such as Velocity.

Response 13: The pseudotime analysis has been performed using Monocle 3¹³. Monocle has been widely used in the single-cell trajectory inference with advantages on bifurcation trajectory prediction¹⁴. Recently, a plethora of trajectory inference methods have been developed but their performances are very variable in different dataset and applying multiple methods can be one good strategy to get more reliable predictions¹⁴. Therefore, we followed the reviewer's suggestion and performed RNA velocity analysis.

We found RNA velocity successfully predicted the future status of individual cell thus adding the direction information in the Monocle analysis.

In the TRIM28^{d/d} epithelium, combined the monocle and RNA velocity results, it is more likely *Aldh1a1* GE is the starting point that differentiate into the other epithelium cells.

In the TRIM28^{ff} epithelium, combined the monocle and RNA velocity results, we found a cell transition from *Lbp* GE toward *Spink1* GE, a branch from *Lgr5* Epi to *Npl* LE and *Lars2* GE.

In the mesenchyme, combined the monocle and RNA velocity results, we are more convinced a cell development route from *A2m* Fibr to *Clec3b* Fibr in TRIM28^{d/d} fibroblasts, from the *Ctla2a* and *Hsd11b2* Fibr toward the *Pcna* and *Mki67* Fibr in TRIM28^{ff} fibroblasts, and a cell transition between *Mustn1* Peri and *Rgs5* Peri. The cell development between pericytes and fibroblasts were only predicted by Monocle.

To avoid bias, we put both monocle and RNA velocity results at Fig S8D, S9C and S9D.

The mesenchyme results were stated at line 386-394.

"We re-clustered all the cells that expressed the mesenchymal makers Pdgfra, Pdgfrb, Acta2, or Vim, including the fibroblast, pericyte and myeloid clusters (Supplementary Fig. 8C) and inferred cell trajectory using Monocle¹³ and RNA velocity¹⁵. As expected, cell transitions within the fibroblast clusters have been observed in the TRIM28^{d/d} (from A2m Fibr to Clec3b Fibr) and TRIM28^{ff} (from Hsd11b2, Ctla2a Fibr toward Pcna, Mki67 Fibr) fibroblasts and pericytes (between Mustn1 and Rgs5 Peri) (Supplementary Fig. 8D). And a transition between the pericytes and fibroblasts have been predicted only by Monocle (Supplementary Fig. 8D) which triggers an interesting cell development proposal but requires further experiments to support."

Supplementary Fig. 8D Mesenchyme

The epithelial results were stated at Line 415-425:

“The single cell trajectory inference showed a possible cell development route from Lars2 GE to Prss29 GE, and a bifurcation from the Lgr5 Epi toward Npl LE and the Lars2 GE in the TRIM28^{ff} epithelium (Supplementary Fig. 9C). It posits a possibility that Lgr5 Epi may be the bipotent stem cells for LE and GE reported before¹⁶. A completely different cell trajectory was predicted for the TRIM28^{dd} epithelium which may start from the Aldh1a1 GE toward the Lars2 GE, or toward the Lcn2 LE and Mki67 Epi (Supplementary Fig. 9D). LGR5 protein was highly expressed in the majority of cells within the TRIM28^{dd} epithelium while only in very few TRIM28^{ff} epithelia (Fig. 8C, Supplementary Fig. 9E). In combination of the increased stem cell marker LGR5 expression and enhanced cell pluripotency pathway in the TRIM28^{dd} epithelium, the TRIM28^{dd} epithelium seems promote the presence of progenitor cells at the expense of more differentiated cells.”

Supplementary Fig. 9C, D Epithelium

Comment 14: Figure 5C legend: again, what's the basis for the blue lines? How are these contacts calculated?

Response 14: "Blue line is the Chromatin loops predicted by HiC" is added in the figure legend

Comment 15: Figure 6A, what is IB? clarify in legend

Response 15: "IB: Immunoblot" was added to the figure legend.

Comment 16: Authors refer to other datasets being used without referencing them in lines 309 and 413.

Response 16:

Now at line 319-320: Reference for “*the published PR and ER α CHIP-Seq from the human endometrium*^{17,18}” were cited;

Now at line 430-431: Reference for “*Indeed, the transcriptomic profile of TRIM28^{d/d} uteri resembles multiple human endometriosis, leiomyoma and endometrial cancer datasets*¹⁹⁻²⁵” were cited.

Comment 17: Correct l. 448 to “condensate” instead of condense.

Response 17: Corrected. Now at line 467.

Comment 18: On l. 326 what do you mean by “pooled”? If they are run on the same lane, how did you differentiate between TRIM28 fl/fl and d/d cells.

Response 18: We pooled three TRIM28^{fl/fl} mice as one TRIM28^{fl/fl} scRNA-seq sample, and then four TRIM28^{d/d} mice as one TRIM28^{d/d} scRNA-seq sample.

The sentence was changed at line 342-343 to “*Three D3.5 TRIM28^{fl/fl} uteri were pooled together then four D3.5 TRIM28^{d/d} uteri were pooled together for scRNA-Seq*”.

Comment 19: RIME results need to be explained. How many replicates, what was the control. Also, full list of proteins identified was not shown (with peptide counts etc) as far as I could see.

Response 19: The HESCs from three different donors were pooled together for RIME analysis. Two technical replicates were generated for PR and IgG RIME. The IgG was used as control. The full list of RIME results was added at Supplementary data 1.

The RIME methods were explained at method line 571-576:

“Primary HESCs from three different donors were decidualized by EPC treatment for three days. The three primary HESCs were fixed by 1% PFA for 8min at RT and mixed as one sample and set for RIME service to Active Motif. The RIME analysis using PR and IgG antibody (see Table S1) were performed twice and IgG is the control. The proteins that were enriched in both two technical replicates were listed as potential PGR interacting proteins.”

Comment 20: Corrections needed on l. 194, l. 235-238, l. 351

Response 20:

Line 194 now 196 corrected as “*TRIM28 deficiency is correlated with suppressed PR and FOXA2 signaling while enhanced ER α signaling (Fig. 3D).*”

Line 235-238 now Line 238-241 corrected as “*Taking the example of Indian hedgehog (Ihh), an epithelial specific PR target gene that plays crucial roles during embryo implantation*²⁶, *Ihh expression was reduced by both TRIM28 deletion and epithelial PR knockout but increased by epithelial PR overexpression in the mouse uterus (Fig. 4C. Supplementary data 3)*”

Line 351 now Line 369-371 corrected as “*Similar to the whole uterus, inflammation IFNG, estrogen, and PR antagonist mifepristone mediated pathways were enriched in the TRIM28^{d/d} fibroblasts.*”

- 1 Wang, X. *et al.* SOX17 regulates uterine epithelial-stromal cross-talk acting via a distal enhancer upstream of *Ihh*. *Nat Commun* **9**, 4421 (2018). <https://doi.org:10.1038/s41467-018-06652-w>
- 2 Rubel, C. A. *et al.* A Gata2-Dependent Transcription Network Regulates Uterine Progesterone Responsiveness and Endometrial Function. *Cell Rep* **17**, 1414-1425 (2016). <https://doi.org:10.1016/j.celrep.2016.09.093>
- 3 Brosens, J. J., Hayashi, N. & White, J. O. Progesterone receptor regulates decidual prolactin expression in differentiating human endometrial stromal cells. *Endocrinology* **140**, 4809-4820 (1999). <https://doi.org:10.1210/endo.140.10.7070>
- 4 Clemm, D. L. *et al.* Differential hormone-dependent phosphorylation of progesterone receptor A and B forms revealed by a phosphoserine site-specific monoclonal antibody. *Mol Endocrinol* **14**, 52-65 (2000). <https://doi.org:10.1210/mend.14.1.0413>
- 5 Nardulli, A. M. & Katzenellenbogen, B. S. Progesterone receptor regulation in T47D human breast cancer cells: analysis by density labeling of progesterone receptor synthesis and degradation and their modulation by progestin. *Endocrinology* **122**, 1532-1540 (1988). <https://doi.org:10.1210/endo-122-4-1532>
- 6 Lionnard, L. *et al.* TRIM17 and TRIM28 antagonistically regulate the ubiquitination and anti-apoptotic activity of BCL2A1. *Cell Death Differ* **26**, 902-917 (2019). <https://doi.org:10.1038/s41418-018-0169-5>
- 7 Taylor, A. H., McParland, P. C., Taylor, D. J. & Bell, S. C. The cytoplasmic 60 kDa progesterone receptor isoform predominates in the human amniochorion and placenta at term. *Reprod Biol Endocrinol* **7**, 22 (2009). <https://doi.org:10.1186/1477-7827-7-22>
- 8 Lee, K. *et al.* Indian hedgehog is a major mediator of progesterone signaling in the mouse uterus. *Nature Genetics* **38**, 1204-1209 (2006). <https://doi.org:10.1038/ng1874>
- 9 Chen, J. R. *et al.* Leukemia inhibitory factor can substitute for nidatory estrogen and is essential to inducing a receptive uterus for implantation but is not essential for subsequent embryogenesis. *Endocrinology* **141**, 4365-4372 (2000). <https://doi.org:DOI> 10.1210/en.141.12.4365
- 10 Mazur, E. C. *et al.* Progesterone receptor transcriptome and cistrome in decidualized human endometrial stromal cells. *Endocrinology* **156**, 2239-2253 (2015). <https://doi.org:10.1210/en.2014-1566>
- 11 Das, S. K. Cell cycle regulatory control for uterine stromal cell decidualization in implantation. *Reproduction* **137**, 889-899 (2009). <https://doi.org:10.1530/REP-08-0539>
- 12 Plaisier, M. Decidualisation and angiogenesis. *Best Pract Res Clin Obstet Gynaecol* **25**, 259-271 (2011). <https://doi.org:10.1016/j.bpobgyn.2010.10.011>
- 13 Cao, J. *et al.* The single-cell transcriptional landscape of mammalian organogenesis. *Nature* **566**, 496-502 (2019). <https://doi.org:10.1038/s41586-019-0969-x>
- 14 Saelens, W., Cannoodt, R., Todorov, H. & Saeys, Y. A comparison of single-cell trajectory inference methods. *Nat Biotechnol* **37**, 547-554 (2019). <https://doi.org:10.1038/s41587-019-0071-9>
- 15 La Manno, G. *et al.* RNA velocity of single cells. *Nature* **560**, 494-498 (2018). <https://doi.org:10.1038/s41586-018-0414-6>

- 16 Jin, S. Y. Bipotent stem cells support the cyclical regeneration of endometrial epithelium of the murine uterus. *P Natl Acad Sci USA* **116**, 6848-6857 (2019).
<https://doi.org:10.1073/pnas.1814597116>
- 17 Chi, R. A. *et al.* Human Endometrial Transcriptome and Progesterone Receptor Cistrome Reveal Important Pathways and Epithelial Regulators. *J Clin Endocrinol Metab* **105**, e1419-1439 (2020).
<https://doi.org:10.1210/clinem/dgz117>
- 18 Hewitt, S. C. *et al.* The Estrogen Receptor alpha Cistrome in Human Endometrium and Epithelial Organoids. *Endocrinology* **163** (2022). <https://doi.org:10.1210/endo/bqac116>
- 19 Herndon, C. N. *et al.* Global Transcriptome Abnormalities of the Eutopic Endometrium From Women With Adenomyosis. *Reprod Sci* **23**, 1289-1303 (2016).
<https://doi.org:10.1177/1933719116650758>
- 20 Pappa, K. I. *et al.* Profiling of Discrete Gynecological Cancers Reveals Novel Transcriptional Modules and Common Features Shared by Other Cancer Types and Embryonic Stem Cells. *PLoS One* **10**, e0142229 (2015). <https://doi.org:10.1371/journal.pone.0142229>
- 21 Day, R. S. & McDade, K. K. A decision theory paradigm for evaluating identifier mapping and filtering methods using data integration. *BMC Bioinformatics* **14**, 223 (2013).
<https://doi.org:10.1186/1471-2105-14-223>
- 22 Hodge, J. C. *et al.* Expression profiling of uterine leiomyomata cytogenetic subgroups reveals distinct signatures in matched myometrium: transcriptional profiling of the t(12;14) and evidence in support of predisposing genetic heterogeneity. *Hum Mol Genet* **21**, 2312-2329 (2012). <https://doi.org:10.1093/hmg/dds051>
- 23 Makinen, N. *et al.* MED12, the Mediator Complex Subunit 12 Gene, Is Mutated at High Frequency in Uterine Leiomyomas. *Science* **334**, 252-255 (2011).
<https://doi.org:10.1126/science.1208930>
- 24 Hendrix, N. D. *et al.* Fibroblast growth factor 9 has oncogenic activity and is a downstream target of Wnt signaling in ovarian endometrioid adenocarcinomas. *Cancer Res* **66**, 1354-1362 (2006).
<https://doi.org:10.1158/0008-5472.Can-05-3694>
- 25 Hawkins, S. M. *et al.* Functional microRNA involved in endometriosis. *Mol Endocrinol* **25**, 821-832 (2011). <https://doi.org:10.1210/me.2010-0371>
- 26 Franco, H. L. *et al.* Ablation of Indian Hedgehog in the Murine Uterus Results in Decreased Cell Cycle Progression, Aberrant Epidermal Growth Factor Signaling, and Increased Estrogen Signaling. *Biology of Reproduction* **82**, 783-790 (2010).
<https://doi.org:10.1095/biolreprod.109.080259>

Reviewer #1 (Remarks to the Author):

The authors have addressed all my concerns. I don't have further comments. congratulations.

Reviewer #2 (Remarks to the Author):

The reviewers have addressed the main concerns by highlighting appropriate replicates used, as well as clarifying analyses and paper presentation.